# Gene regulatory programs underlying diversification of facial ligaments and tendons in zebrafish

Ryan R. Roberts[1], Arshia Bhojwani[1,2], Kuo-Chang Tseng[1], Kelsey Elliott[1], Hung-Jhen Chen[1], Lauren Teubner[1], Desmarie Sherwood[3], Joanna Smeeton[1,3], Cameron L. Miller[4], Pavan K. Nayak[4], Arul Subramanian[4], Thomas F. Schilling[4], Amy E. Merrill[1,2] and J. Gage Crump[1,*]

## ABSTRACT

Whereas ligaments hold skeletal elements together, tendons bridge the musculature with the skeleton. How connective tissues of the right type and function are specified in distinct regions of the developing body remains unclear. Here, we have generated single-cell datasets of RNA expression and chromatin accessibility for *scxa:mCherry*+ connective tissues of the developing zebrafish face. We identified cell clusters corresponding to tendon, ligament, periligament, perichondrium and other types, as well as tendon and ligament subtypes with an osteogenic signature that may explain the remodeling of ligament-bone interfaces and the formation of sesamoid bones. We further identified several enhancers driving spatially restricted transgenic activity in ligaments, periligament tissue and other connective tissues. By utilizing a ligament-specific photoconvertible nlsEOS transgenic line, we revealed directional growth of ligaments. In addition, we found that *nkx3.2* is expressed within the joint-proximal domain of the major jaw-stabilizing ligament, with this domain being lost in *nkx3.2* mutants. Our study reveals distinct gene regulatory programs for jaw connective tissue diversification and provides a mechanism underlying the propensity of tendons and ligaments to ossify in normal and pathological contexts.

KEY WORDS: Single-cell genomics, Ligament, Tendon, Zebrafish, Jaw, Nkx3.2

## INTRODUCTION

The precise integration of muscles, bones and cartilages by tendons and ligaments facilitates movement of the vertebrate body. A typical tendon transduces mechanical force generated by a muscle to the graded enthesis attaching it to a cartilaginous or bony element, while others connect adjacent muscles. Tendons can also be classified into longer force-transmitting and shorter muscle-anchoring varieties. Ligaments resemble tendons but function to stabilize the skeleton by interconnecting bones and cartilages. While most tendons and ligaments in the head derive from cranial neural crest cells (CNCCs), they are derived from paraxial and lateral plate mesoderm in the trunk and limbs, respectively (Durland et al., 2008; Evans and Noden, 2006; Kontges and Lumsden, 1996). Ligaments and tendons share an extracellular matrix (ECM) rich in collagen type I (Col1) fibers. While these fibers tend to lie in parallel bundles in tendons, reflecting the direction of muscle contractility, ligaments often have a more disorganized lattice fiber structure (Amiel et al., 1984; Benjamin et al., 2008). Both are sites of overuse injuries that often fail to repair without surgical intervention. A better understanding of the unique developmental trajectories of tendons and ligaments would aid in future approaches to regenerating the correct types of connective tissues.

Ligaments and tendons derive from progenitor cells expressing the basic helix-loop-helix transcription factor Scleraxis (Scx) (Cserjesi et al., 1995; Schweitzer et al., 2001). However, deletion of *Scx* in mice results in relatively mild connective tissue defects, affecting primarily the longer force-transmitting tendons (Murchison et al., 2007). Similarly, in zebrafish, combined loss of *scxa* and *scxb* results in a reduction but not complete loss of tendons and ligaments throughout the body (Kague et al., 2019). The Mohawk (Mkx) transcription factor is also required for tendon development, with *Mkx*[−/−] mice displaying hypoplastic tendons but not reductions in tendon cell numbers (Ito et al., 2010). In the fish and mouse jaw, the orphan nuclear receptor Nr5a2 is required for tendon and ligament formation, in part by directly activating *scxa* enhancers (Chen et al., 2023). In addition, a recent study has identified members of the CREB and EBF transcription factor families as sufficient to induce tendon fates in zebrafish and human cells, although genetic deletion of *ebf1a* and *ebf3a* in zebrafish resulted in only minor tendon defects (Niu et al., 2024). Given that none of these mutants has complete loss of tendons and ligaments, it is clear that gaps remain in our understanding of the developmental gene regulatory mechanisms specifying tendons and ligaments. Moreover, though there have been studies on directional tendon elongation, little is known about how individual tendons and ligaments remodel and grow as the body develops.

Tendons and ligaments share Col1 and other ECM components with bone, yet they also have distinct ECM components such as Tenomodulin (Tnmd) and Thrombospondin 4 (Thbs4) (Jelinsky et al., 2010; Shukunami et al., 2018; Subramanian and Schilling, 2014). Overlapping ECM compositions of bone, tendons and ligaments could reflect convergence of gene expression after differentiation and/or the origin of these cell types from common precursors. Tendons and ligaments can also undergo progressive ossification with age, or in response to trauma in some cases of heterotopic ossification (Meyers et al., 2019). There are also congenital anomalies of inappropriate connective tissue ossification, such as calcification of the stylohyoid ligament in Eagle Syndrome (Camarda et al., 1989). In some cases,

[1]Department of Stem Cell Biology and Regenerative Medicine, Keck School of Medicine, University of Southern California, Los Angeles, CA 90033, USA. [2]Center for Craniofacial Molecular Biology, Ostrow School of Dentistry, University of Southern California, Los Angeles, CA 90033, USA. [3]Department of Rehabilitation and Regenerative Medicine, Columbia Stem Cell Initiative, Columbia University Irving Medical Center, New York, NY 10032, USA. [4]Department of Developmental and Cell Biology, University of California, Irvine 92697, USA.

*Author for correspondence (gcrump@usc.edu)

C.L.M., 0009-0005-7407-0698; P.K.N., 0000-0002-4360-6729; A.S., 0000-0001-8455-6804; T.F.S., 0000-0003-1798-8695; J.G.C., 0000-0002-3209-0026

loss of key factors for tendon and ligament development may result in connective tissue cells progressively adopting a skeletal fate, with, for example, $Mkx^{-/-}$ mice displaying ossification of the Achilles tendon at postnatal stages (Liu et al., 2019).

To understand the developmental diversification of connective tissues, we generated single-cell datasets of gene expression and chromatin accessibility for $scxa:mCherry^+$ cells of the larval zebrafish head. We identified distinct tendon and ligament clusters, as well as a cluster with mixed tendon/ligament and osteoblast identity. We also identified a perilligament cluster and two distinct perichondrium clusters. Analysis of transcription factor motifs enriched in open chromatin showed that tendon and ligament clusters are co-enriched for Scx and Runx2 motifs, consistent with the role of Runx2 in osteogenic gene expression (Ducy et al., 1997) and the tendency of some tendons and ligaments to ossify. Consistently, we found that a subset of cells within the major jaw joint-associated ligament initiates expression of an $sp7:GFP$ osteoblast reporter at juvenile stages, which coincides with remodeling of the ligament to connect to late-developing bones. We also observed co-expression of $scxa:mCherry$ and osteoblast transgenes in the muscle-anchoring tendons of the branchiostegal ray (br) bones, and in a midline tendon that forms a sesamoid bone. By testing open chromatin regions in transgenic assays, we uncovered enhancers active in subsets of ligaments, perilligament cells and perichondrium. Using one of these ligament-specific enhancers to drive expression of the photoconvertible nlsEOS fluorescent protein, we demonstrated that ligament cell addition occurs in a dynamic and directional fashion. At the anterior end of the major jaw-stabilizing ligament, mutant analysis revealed requirements for the transcription factor Nkx3.2 in integration of the ligament with the jaw joint. Collectively, these findings reveal heterogeneity in developmental specification and maturation between and within individual ligaments and tendons.

## RESULTS

### Late-emerging ligaments and tendons in the zebrafish head

We first set out to catalog the diversity of tendons and ligaments in the zebrafish head, in particular those forming during later larval and juvenile stages that have been less studied. To do so, we used the $scxa:mCherry$ transgene that labels facial mesenchyme at ~2 days post-fertilization (dpf) and becomes restricted to ligaments and tendons over the next few days (Anderson et al., 2023; Chen et al., 2023; Chen and Galloway, 2014). We performed imaging of $scxa:mCherry^+$ cells at 5 and 14 dpf in relation to either $col2a1a:GFP^+$ chondrocytes and phalloidin-stained muscle, or to $sp7:GFP^+$ osteoblasts (Fig. 1; Fig. S1). Here, we introduce a revised naming system for tendons and ligaments that defines which muscles, cartilages and/or bones they connect (Fig. 2; Subramanian et al., 2026). Examples of late-developing structures include ch-br ligaments connecting the series of branchiostegal ray (br) bones to the ceratohyal (ch) cartilage, and br-HH tendons lining each br bone and connecting to thin hyohyal (HH) muscle fibers.

### Single-cell analysis of zebrafish facial connective tissues

To define gene expression and open chromatin within facial tendon, ligament and associated mesenchyme cells, we isolated $scxa:mCherry^+$ cells from the developing zebrafish face and performed single-cell RNA sequencing (scRNA-seq) at 5 dpf (1491 cells) and 14 dpf (17,220 cells), single-nuclei ATAC sequencing (snATAC-seq) at 5 dpf (3927 cells), and combined single-nuclei RNA/ATAC sequencing (multiome) at 5 dpf (8665 cells) using the 10X

Genomics platform, Illumina sequencing and filtering for quality ($n$=1 for each, see Materials and Methods for details of animal pooling and workflow). After converting the snATAC-seq library to pseudo-RNA expression using the SnapATAC algorithm (Fang et al., 2021), we integrated the four datasets for subsequent analysis (Fig. 3A; Fig. S2A; Table S1). As this integrated dataset contained some epithelial, blood, neuronal and other non-mesenchymal cells, we used expression of $col2a1a$ (chondrocytes), $sp7$ (osteoblasts) and $thbs4b$ (soft connective tissue) to enrich for connective tissue types (Fig. 3B-D; Fig. S2B,C; Fig. S3A; Table S1). After re-clustering of the connective tissue subset, we identified two clusters of osteoblasts ($ifitm5^+$), four clusters of chondrocytes ($sox9a^+$), including one articular ($f13a1b^+$), two clusters of tendon and ligament cells ($scxa$-high, $thbs4b$-high), five $thbs4b^+$ connective tissue clusters of unclear identity, and one cluster each of mesenchyme ($cilp^+$, $scxa$-low, $thbs4b$-low), perichondrium ($hyal4^+$; Fabian et al., 2022), dermal fibroblasts ($pah^+$) and proliferative cells. Despite filtering for connective tissue cells, we also found four unknown clusters enriched for DNA damage pathways that may represent cells damaged during isolation.

To further uncover heterogeneity within tendon, ligament and associated mesenchyme clusters, we combined tendon/ligament, osteoblast, chondrocyte, perichondrium and mesenchyme clusters from the connective tissue subset and performed a second round of re-clustering to generate 16 new clusters (Fig. 4A,B; Fig. S2D; Table S1). Based on previous data and in situ hybridization validation below, these include ligamentocytes [$thbs4a^+$ (Anderson et al., 2023), $ognb^+$ and $mkxa^+$], tenocytes ($tnn^+$ and $cilp2^+$), 'ossifying tendon/ligament' cells ($angptl2b^+$) based on higher expression of osteoblast-related genes $sp7$ and $ifitm5$, perilligament cells ($cilp$-high; $thbs4b$-low), distinct $hyal4^+$ and $spon2b^+$ perichondrium cells, mesenchyme cells ($alx4a^+$, $meis1b^+$), three clusters of chondrocytes, four clusters of osteoblasts, and two clusters of unknown and potentially damaged cells. When analyzed for $scxa$ expression, all clusters in the connective tissue subset and tendon and ligament subset had variable numbers of $scxa^+$ cells, while some epithelia and red blood cell clusters in the original combined datasets had no $scxa^+$ cells, suggesting recovery from FACS due to autofluorescence (Fig. S2E). In addition, similar proportion of $scxa^+$ cells in some osteoblast and chondrocyte clusters compared to tendon and ligament clusters is consistent with these representing enthesis-like clusters intermediate between skeletal and soft connective tissue types.

### RNA in situ validation of connective tissue cell types

To validate the clusters identified by single-cell analyses, we performed RNA in situ hybridization in zebrafish larvae using highly sensitive hybridization chain reaction (HCR) and RNAscope methodologies with dual or triple RNA probes, or combining RNA probes with an anti-mCherry antibody to detect $scxa:mCherry^+$ tendons and ligaments (Fig. 4C,D; Fig. S3B-E). We observed that $tnmd$ was a broad marker of tendons and ligaments. For the tenocyte cluster, we observed expression of $tnn$ and $cilp2$ in partially overlapping subsets of facial tendons. For the ligamentocyte cluster, $thbs4a$, $ognb$ and $mkxa$ expression were co-expressed in the mc-op ligament at 3 and 5 dpf and the ra-iop ligament at 14 dpf. Similarly, $ucmaa$ and $ecrg4a$ were expressed in mc-op, but also other domains, at 5 dpf. We also observed expression of $thbs4a$ in the series of ch-br ligaments and a bilateral pair of long midline ch-HH tendons, suggesting that $thbs4a$ is a pan-ligament marker with some additional expression in a subset of long, force-transmitting tendons. A time-course of double in situ hybridization expression at 3, 5 and 14 dpf

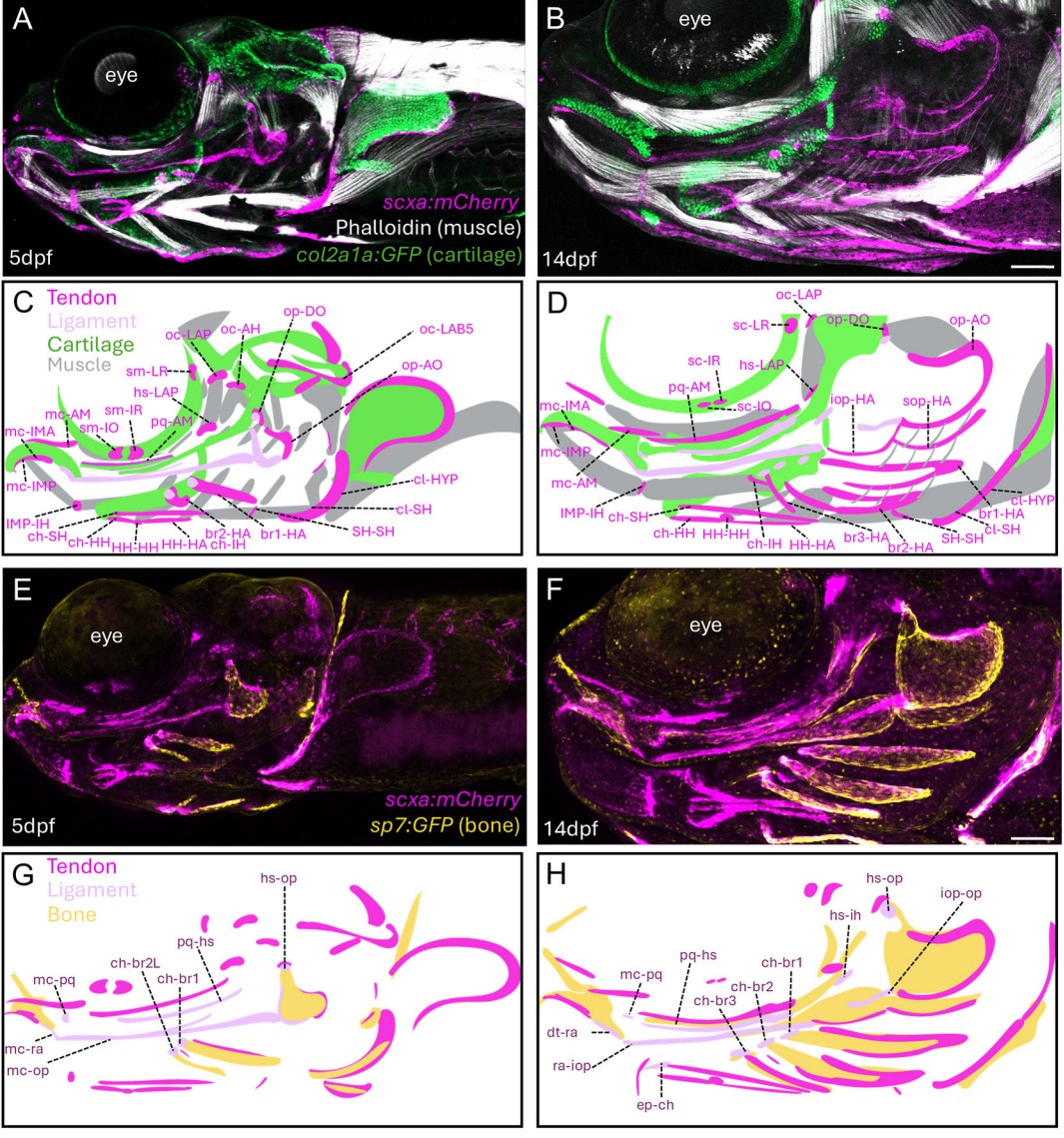

**Fig. 1. Tendons and ligaments of the developing zebrafish face.** (A,B) Confocal projections in lateral view show facial tendons and ligaments (*scxa: mCherry*), muscles (Phalloidin) and cartilages (*col2a1a:GFP*) at 5 dpf (A; *n*=3) and 14 dpf (B; *n*=3). (C,D) Schematics with tendon abbreviations in magenta. (E,F) Confocal projections in lateral view show facial tendons and ligaments (*scxa:mCherry*), and bones (*sp7:GFP*) at 5 dpf (E; *n*=3) and 14 dpf (F; *n*=3). (G,H) Schematics with ligament abbreviations in lavender. Scale bars: 100 μm. See Fig. 2 for complete list of names and abbreviations.

confirmed mutually exclusive expression of *thbs4a* in ligaments and the long ch-HH tendons, versus *cilp2* in most other tendons (Fig. 4E). For markers of the ossifying tendon and ligament cluster, *angptl2b* was expressed in a number of facial tendons and ligaments, including the region of the mc-op ligament that ossifies and fuses to the iop bone, the br-HA, sop-HA and iop-HA tendons that express the osteoblast transgene *sp7:GFP*, and the ch-SH tendons that form a sesamoid bone. For the periligament cluster, we observed *cilp* expression in *scxa:mCherry*-negative mesenchyme cells surrounding mc-op. In addition to tenocyte expression, *cilp2* was also expressed in this periligament mesenchyme. We also observed *hyal4* and *spon2b* expression in distinct subsets of perichondrium cells, which we confirmed by double *in situ* hybridization. Lastly, *meis1b* and *abi3bpa* were broad markers of undifferentiated mesenchyme, with triple in situs showing that *meis1b* overlaps with *thbs4a* in ligamentocytes but not with *cilp* in periligament cells. *In situ* validation thus confirms most clusters while revealing additional heterogeneity within tendon and ligament clusters.

## Distinct enhancer architecture of tendon and ligament subtypes

We next queried our snATACseq datasets for potential differences in chromatin architecture between the three tendon and ligament clusters. To identify transcription factor-binding motifs enriched in each cluster, we performed *de novo* motif enrichment for each cluster versus random zebrafish genomic sequence using Hypergeometric Optimization of Motif EnRichment (HOMER) (Fig. 5A). The top motif for each cluster was a bHLH motif (GCCATCTAGTGG) matching that reported for mouse Scx using chromatin immunoprecipitation and sequencing (Liu et al., 2021), consistent with the key role of this factor in tendon and ligament development (Murchison et al., 2007). Within the top three motifs for each tendon and ligament cluster was also an Nfat motif [GGAAA(A/T)] nearly identical to the second most common motif enriched in Scx-bound regions in the developing mouse forelimb (Liu et al., 2021). These similarities suggest that Scx and Nfat family members may broadly regulate tendon and

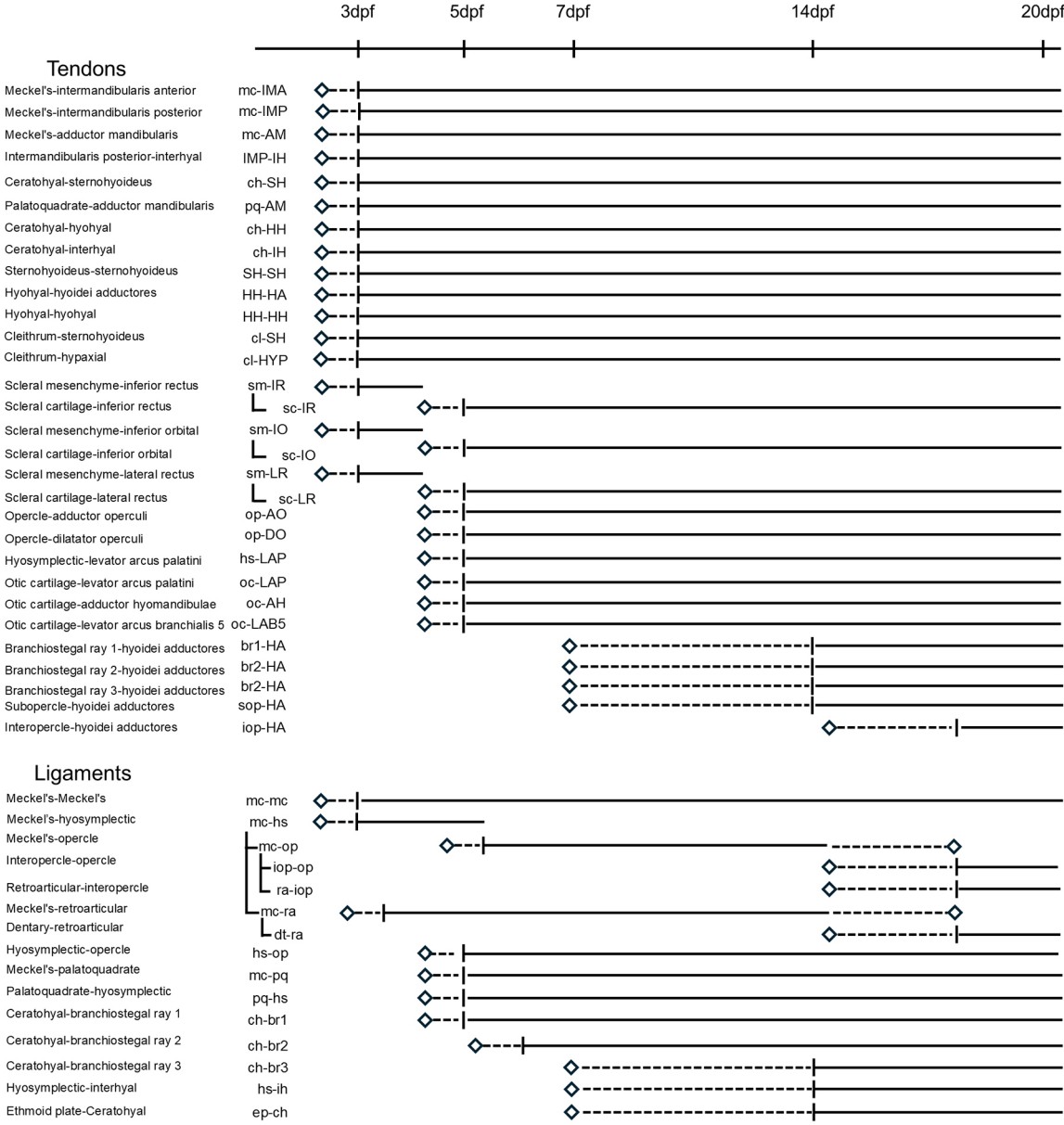

**Fig. 2. Timeline of tendon and ligament development.** Summary of the emergence of facial tendons and ligaments in the zebrafish face. Abbreviations indicate connection points with skeletal elements (lowercase) and muscles (uppercase).

ligament development across the body in both fish and mammals. Unexpectedly, a motif closely matching mouse Runx2 (AACCACA) (Wu et al., 2014) was the 2nd highest for the tendon cluster, and 7th highest for the ligament and ossifying tendon and ligament cluster. When we used Signac to further refine differentially enriched motifs per cluster, by comparing against a randomized background of 40,000 motifs from just the *scxa: mCherry*+ tendon and ligament subset, RUNX motifs were selectively enriched in only the ossifying tendon and ligament cluster (Fig. S4).

Given the essential role of Runx2 in osteoblast differentiation (Ducy et al., 1997), we asked whether enrichment of RUNX motifs reflected cells with mixed bone and tendon/ligament identity. Consistently, we observed co-localization of *scxa* with the bone marker *sp7* in ch-br1 at 5 dpf, and co-localization of *scxa:mCherry* and *sp7:GFP* in the series of br-HA tendons connecting the fine muscle fibers and br bones at 14 dpf (Fig. 5B,C). Calcein staining of

the br2 bone revealed that many of these *scxa:mCherry*+; *sp7:GFP*+ cells were embedded in mineralized matrix, with a single-cell layer of non-mineralized *scxa:mCherry*+ cells on the surface of the bone (Fig. 5D). We also observed *scxa:mCherry*+; *sp7:GFP*+ cells associated with the newly formed iop bone at 20 dpf, which corresponds to the region where the initial mc-op ligament had split into distinct ra-iop and iop-op ligaments (Fig. 5E,F). This suggests that a subset of ligament cells may ossify and fuse to the late-developing iop bone to remodel ligament attachments. At the midline, the ch-SH tendons were also double positive for *scxa: mCherry* and the osteoblast transgenes *RUNX2:GFP* and *sp7:GFP* at 14 dpf, with Calcein staining revealing mineralization indicative of a sesamoid bone (Fig. 5G-I). These findings support our single-cell analysis that there is a cell type with mixed tendon/ligament and osteoblast identity that is present at diverse locations, including tendon and ligament attachments to bone, areas of ligament remodeling and in tendon-derived sesamoid bones.

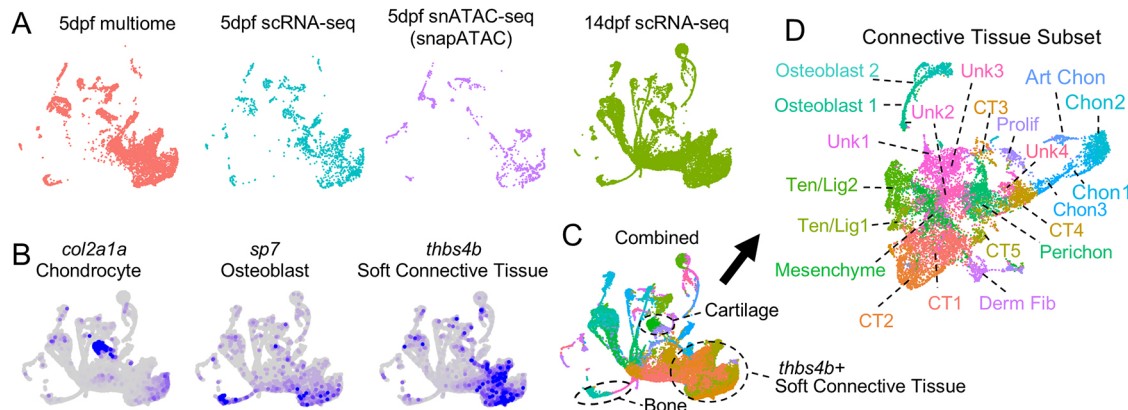

**Fig. 3. Single-cell datasets of facial connective tissues.** (A) Uniform manifold approximation and projection (UMAP) combining the four datasets generated from *scxa:mCherry*⁺ cells sorted from zebrafish heads. Each library is shown individually. To integrate libraries, the RNA component of the 5 dpf single-nuclei multiome experiment was used, and the snapATAC algorithm was used to create a pseudo-RNA library for the 5 dpf snATAC-seq dataset. (B) Feature plots of integrated datasets show markers for chondrocytes (*col2a1a*), osteoblasts (*sp7*) and soft connective tissue (*thbs4b*). (C) UMAP shows integrated datasets, with the indicated cartilage, bone and *thbs4b*⁺ soft connective tissue clusters used for re-clustering. (D) Re-clustering of cartilage, bone and soft connective tissue clusters with cluster names indicated. Art Chon, articular chondrocyte; Chon, chondrocyte; CT, connective tissue; Derm Fib, dermal fibroblast; Perichon, perichondrium; Prolif, proliferative; Ten/Lig, tendon and ligament; Unk, unknown.

## Identification of enhancers active in connective tissue subtypes

Analysis of cluster-specific chromatin accessibility in snATAC-seq datasets revealed a number of differentially accessible regions (DARs) near genes that we found to be expressed in connective tissue subtypes and to have previously reported roles in connective tissue and/or skeletal development (Fig. 6A). We therefore tested whether these DARs could drive transgene activity at 5-8 dpf in combination with an E1B minimal promoter. We confirmed expression in multiple founder lines for each enhancer (Fig. S5), with the exception of *ucmaa_p1:GFP* for which only a single founder was identified. DARs linked to *thbs4a*, *sparc* and *ucmaa* drove nlsEOS or GFP expression in the mc-op ligament, with *thbs4a_p1:nlsEOS* more broadly marking ligaments such as ch-br1 (Fig. 6B-E). DARs for *cilp*, *scxa* and *fgfrl1a* drove periligament expression surrounding mc-op, with the *fgfrl1a* DAR also having activity in joint-associated perichondrium (Fig. 6F-H). The *scxa_p3* DAR is different from two previously reported DARs that are active in jaw tendons and directly regulated by Nr5a2 (Chen et al., 2023), although in a second *scxa_p3:GFP* founder, expression was higher in chondrocytes than in periligament cells. We also identified DARs near *six2a* and *meox1* that drove expression in perichondrium, in particular associated with joints (Fig. 6I,J). While these DARs were all accessible in the clusters in which they drove transgenic activity, they also showed variable accessibility in other clusters, showing that accessibility alone does not control activity for many connective tissue enhancers. The high degree of specificity of many enhancers also reveals gene regulatory heterogeneity for connective tissue subtypes beyond that revealed by single-cell sequencing-based clustering alone.

## Photoconversion-based lineage tracing reveals directional growth of facial ligaments

How ligaments grow during development remains poorly understood. We therefore took advantage of our newly developed *thbs4a_p1:nlsEOS* line to track the addition of new ligamentocytes during growth of the face. Exposure to UV light permanently photoconverts green nlsEOS protein to red fluorescence, and the long stability of nlsEOS protein allows photoconverted cells to be traced over days and sometimes weeks of development (Fabian et al., 2022; Farmer et al., 2024). Cells that newly express the transgene after

photoconversion would appear only green. To understand dynamics of ligament growth, we performed photoconversion and re-imaging of *thbs4a_p1:nlsEOS*⁺ ligament cells from 3 to 5, 5 to 7, 7 to 14, and 14 to 20 dpf, and quantified the red-to-green ratio of nlsEOS fluorescence along the anterior (jaw joint) to posterior (iop) axis (Fig. 7; Fig. S6). We observed a higher green-to-red nlsEOS ratio indicative of new ligamentocyte addition toward the posterior end of the mc-op ligament for all traces except 14 to 20 dpf. From 7 to 14 dpf and 14 to 20 dpf, we observed a shift where new green-only cells were added at more medial rather than more-posterior positions. In addition, no new green fluorescence was seen in a small group of cells where the ligament connects to the joint for any of the traces, indicating that the jaw joint attachment cells form prior to 3 dpf. We also observed directional growth at the hs-op and series of ch-br ligaments, with new ligamentocytes added at the bone sides. The hs-op ligament undergoes cell addition largely from 3 to 5 dpf, ch-br1 largely from 3 to 5 dpf and 5 to 7 dpf, and ch-br2 from 7 to 14 dpf and 14 to 20 hpf, corresponding, respectively, to the progressive appearance of the op, br1 and br2 bones during development. These lineage-tracing experiments reveal dynamic and directional growth of facial ligaments that coincides with the progressive appearance of the opercular series of intramembranous bones.

## Nkx3.2 is required for development of the anterior region of the mc-op ligament

Given that the mc-op ligament grows through preferential cell addition at posterior then medial regions, and that a small group of jaw joint-associated ligament cells are specified early, we asked whether the anterior-most region of mc-op might form through distinct mechanisms. Embryonic expression of *nkx3.2* marks the jaw joint and surrounding tissues, and *nkx3.2* mutants fail to develop a jaw joint and the RA bone to which mc-op connects (Miller et al., 2003; Smeeton et al., 2021). We therefore examined *nkx3.2* expression and found that it colocalized with *thbs4a* in the jaw joint-proximal anterior region of the mc-op ligament at 5 dpf (Fig. 8A). In *nkx3.2⁻/⁻* mutants, *thbs4a* expression was disorganized and *ognb* expression was lost in the anterior region of the mc-op at 3 dpf (Fig. 8B). We also observed reduced levels of the ligament transgene *thbs4a_p1:nlsEOS* and a dysmorphic domain (single layer instead of two) of the periligament transgene

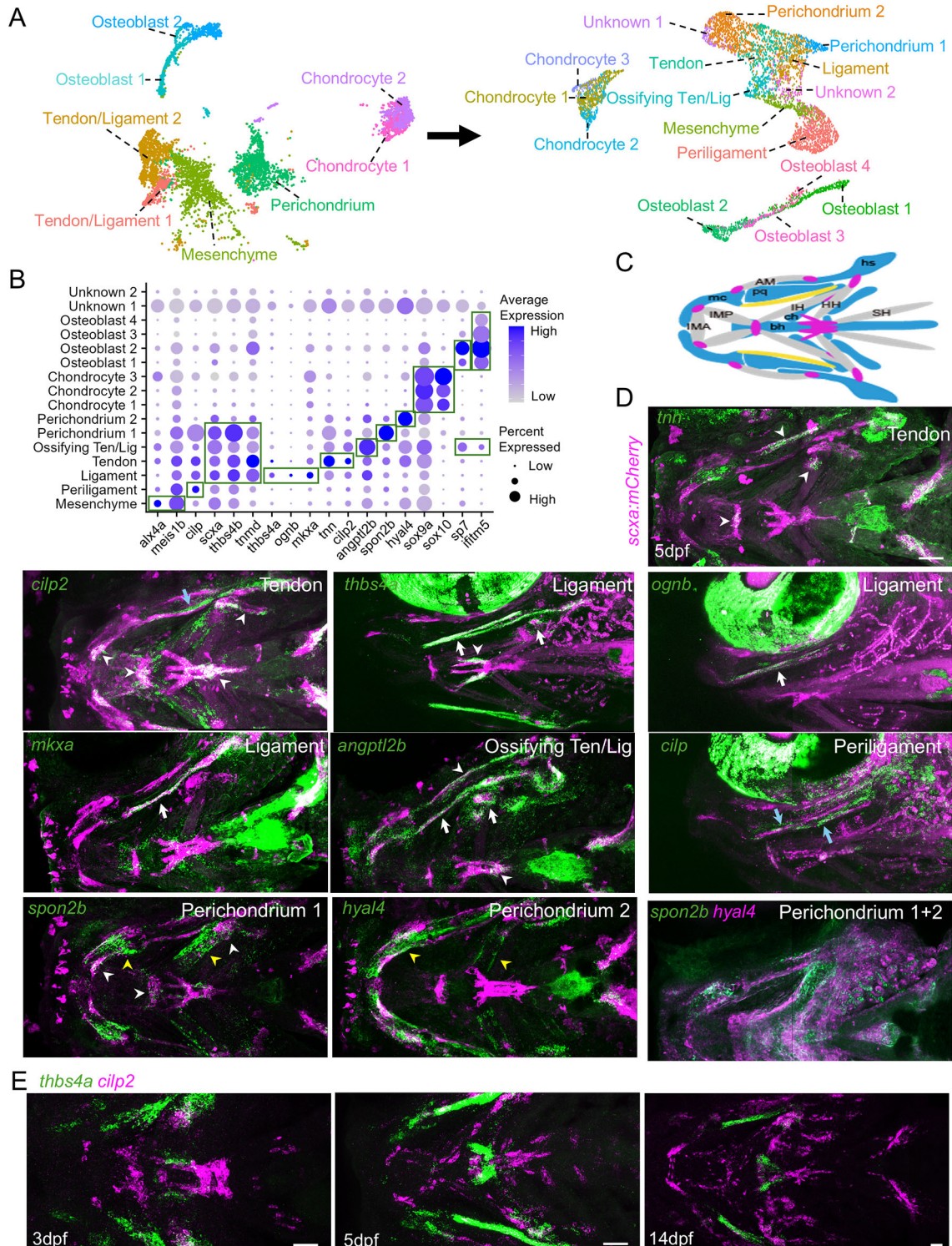

Fig. 4. Connective tissue gene expression in the zebrafish face. (A) Clusters of the connective tissue subset from Fig. 3D (left) used for re-clustering to generate the tendon and ligament subset (right). (B) Dot plot shows select marker genes for each cluster. Green outlines highlight genes with higher expression in the corresponding clusters. (C) Schematic diagram of the ventral view of the zebrafish head, highlighting muscles (gray) and cartilages (blue) for reference. Cartilages: bh, basihyal; ch, ceratohyal; hs, hyosymplectic; mc, Meckel's; pq, palatoquadrate. Muscles: AM, adductor mandibularis; HH, hyohyal; IH, interhyoideus; IMA, intermandibularis anterior; IMP, intermandibularis posterior; SH, sternohyoideus. (D) RNAscope *in situ* hybridization, except for HCR *in situ* hybridization for *thbs4a*, *ognb* and *cilp*, for indicated genes (green). Confocal projections of the 5 dpf face are shown with expression noted for ligaments (white arrows), tendons (white arrowheads), periligament (blue arrows) and perichondrium (yellow arrowheads). Except for *hyal4* and *spon2b* double *in situ*, anti-mCherry staining of *scxa:mCherry* labels tendons and ligaments (magenta). *thbs4a*, *n*=9; *ognb*, *n*=9; *mkxa*, *n*=6; *angptl2b*, *n*=3; *cilp*, *n*=7; *spon2b*, *n*=10; *hyal4*, *n*=4; *hyal4*/*spon2b*, *n*=6. (E) RNAscope *in situ* hybridization shows largely non-overlapping expression of *thbs4a* and *cilp2* in the zebrafish face at 3, 5 and 14 dpf (*n*=3 each stage). Scale bars: 50 µm.

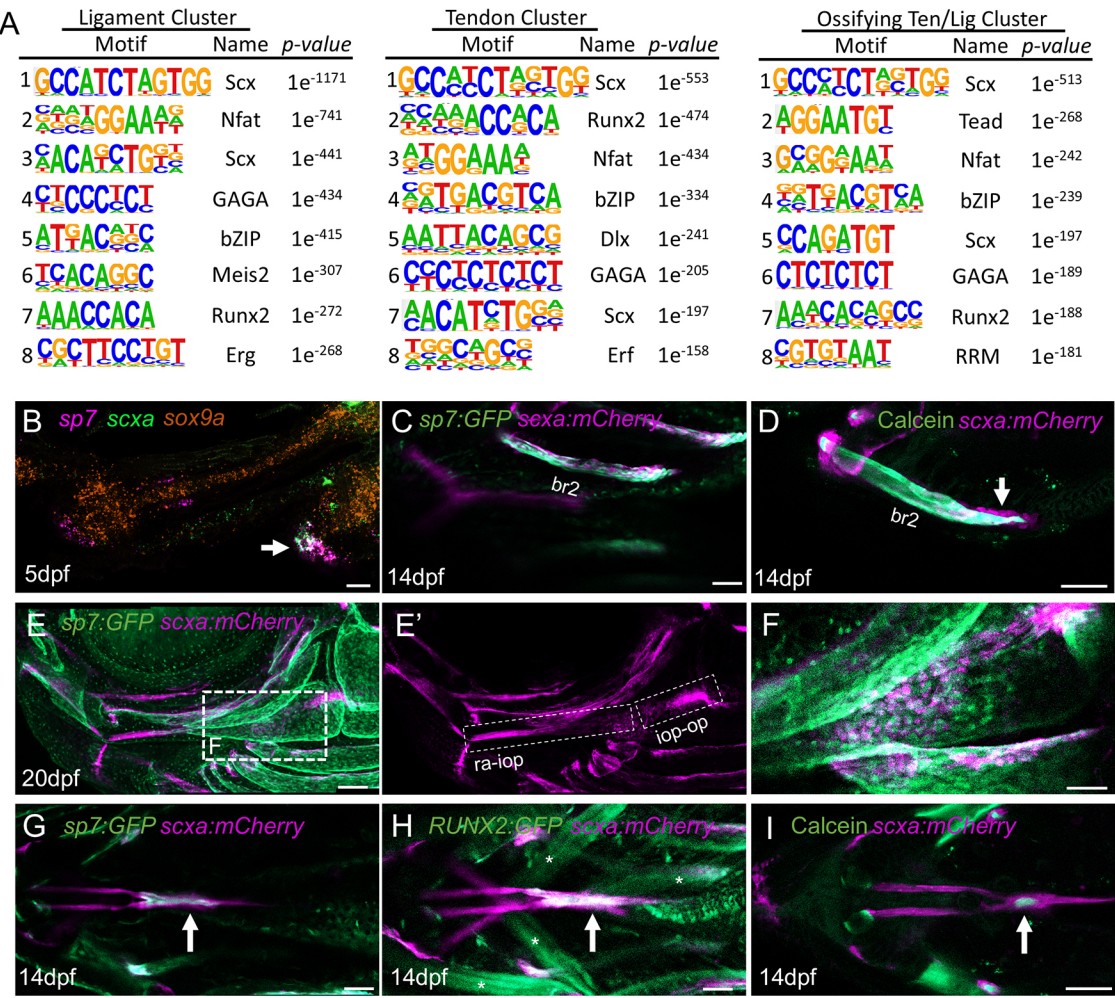

**Fig. 5. Motif enrichment and osteogenesis of tendon and ligament subtypes.** (A) HOMER *de novo* motif enrichment identifies the top 8 motifs for the ligament, tendon, and ossifying tendon and ligament clusters. (B) Single confocal section of RNAscope *in situ* of the 5 dpf face in lateral view shows expression of *sp7* (magenta), *scxa* (green) and *sox9a* (orange). Arrow indicates co-expression of *sp7* and *scxa* at the attachment site of the br1 bone to the ch-br1 ligament (*n*=3). (C,D) Single confocal section shows co-expression of *sp7:GFP* and *scxa:mCherry* in cells of the br2 bone (C). Calcein staining shows the presence of *scxa:mCherry*⁺ cells within the mineralized matrix, as well as a single row of *scxa:mCherry*⁺ cells (arrow) on the surface of the mineralized bone (D) (*n*=3 each). (E-F) Confocal projections of the 20 dpf face in lateral view show expression of *sp7:GFP* and *scxa:mCherry*. (E′) The outlined regions in the *scxa:mCherry* channel show distinct mc-iop and iop-op ligaments. (F) A higher magnification single confocal section corresponding to the region outlined in E shows colocalization of *sp7:GFP* and *scxa:mCherry* in the region of the iop bone (*n*=3). (G-I) Single confocal sections of the 14 dpf face in ventral view show colocalization of *scxa:mCherry* with *sp7:GFP* (G), *RUNX2:GFP* (H) and Calcein⁺ mineralized matrix (I) in a sesamoid bone (arrows) within the ch-SH tendons (*n*=3 each). Asterisks in H indicate weak off-target expression of *RUNX2:GFP* in cranial muscles. Scale bars: 20 μm in B; 50 μm in C-E′, G-I; 100 μm in F.

*cilp_p1:nlsEOS* in the anterior mc-op region at 5 dpf (Fig. 8C). At 6 dpf, the mc-op ligament expressed higher levels of Col1a1a protein versus other facial tendons and ligaments, and we found a localized reduction of Col1a1a in the anterior domain in mutants (Fig. 8D). These data show that Nkx3.2 is selectively required for the development of the anterior-most domain of mc-op that connects to the jaw joint.

## DISCUSSION

Tendons and ligaments are highly related connective tissue types that have been hard to distinguish by gene expression alone (Bobzin et al., 2021). By globally comparing both gene expression and chromatin accessibility among diverse tendons and ligaments of the zebrafish face, we have uncovered gene expression and enhancer architectures distinguishing these two major connective tissue types. We also found that a subset of tendon and ligament cells express osteogenic genes, which may aid in forming bone attachments, remodeling ligaments as new bones develop and creating tendon-associated sesamoid bones. In addition, we uncovered distinct regulatory mechanisms within domains of individual ligaments. These findings reveal heterogeneity both between and within individual tendons and ligaments that may help specify their unique structural properties, tendency to ossify and ability to remodel as they grow in concert with the skeleton.

Although very few genes show exclusive expression in tendons versus ligaments, we did find many genes with differential expression. For example, ligaments express higher levels of *thbs4a*, *ognb*, *mkxa* and *ucmaa*, and tendons express higher levels of *tnn* and *cilp2*. The major type 1 collagen encoded by *col1a1a* is also higher in zebrafish facial ligaments than tendons, consistent with ECM differences observed in tendon and ligament subtypes at earlier stages (Subramanian et al., 2026). Zebrafish have two orthologs of Thbs4,

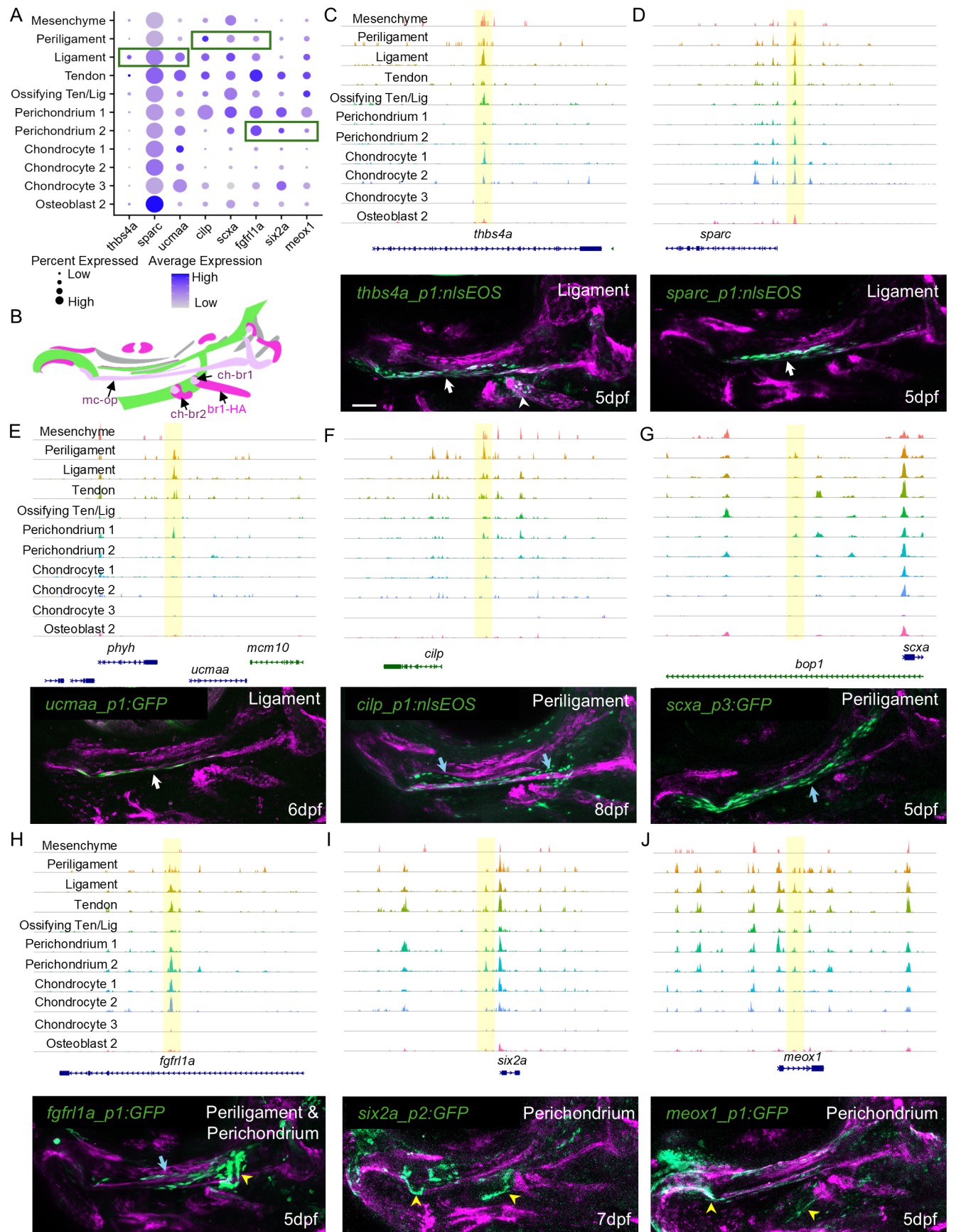

**Fig. 6.** See next page for legend.

**Fig. 6. Transgenic activity of connective tissue enhancers.** (A) Dot plots show expression of genes linked to tested differentially accessible regions (DARs) in selected connective tissue clusters from Fig. 4A. Green outlines correspond to cell types displaying transgenic activity of tested enhancers linked to the listed genes. (B) Schematic of the 5 dpf face in lateral view showing major ligaments (pink) and tendons (magenta) labeled by transgenes, with cartilages shown in green and muscles shown in gray for reference. (C-J) Genome coverage plots show chromatin accessibility from snATAC-seq datasets in the indicated clusters. Enhancers tested in stable transgenic lines are highlighted in yellow. Confocal projections of the zebrafish face in lateral view are shown for each enhancer transgenic (green) with *scxa:mCherry* (magenta) labeling tendons and ligaments. Annotations indicate ligaments (white arrows), periligament (blue arrows) and perichondrium (yellow arrowheads) (*n*=3 each). Scale bar: 50 µm.

and *thbs4b* RNA and protein broadly mark tendons and ligaments in zebrafish (Subramanian and Schilling, 2014). Selective expression of *thbs4a* in ligaments could serve to generate higher overall levels of Thbs4 protein compared to tendons, which could influence the unique ECM and hence structural properties of these two connective tissue types. Compared to our 3 dpf scRNA-seq analysis that resolved many more clusters of tendon and ligaments cells (Subramanian et al., 2026), analysis at 5 and 14 dpf resolved only three major types. This was not simply due to the resolution of clustering, as further clustering did not reveal distinct sub-clusters of tendon and ligament cells that we could validate by *in situ* hybridization. One possible explanation for reduced cluster numbers at later stages is the persistence of regional patterning gene expression at the 3 dpf time-point (Fabian et al., 2022), which, in combination with connective tissue gene expression, would help to distinguish individual tendons and ligaments. At the later 5 and 14 dpf time-points, earlier regional gene expression has substantially decreased, thus potentially forcing tendon and ligament cells to coalesce into fewer clusters. Nonetheless, in our *in situ* validation of marker genes, we did observe extensive heterogeneity of gene expression reflecting the varied structural properties of

individual tendons and ligaments observed at 3 dpf by Subramanian et al. (2026).

Differential gene expression in ligaments versus tendons was also reflected by the identification of several enhancers that are specific for ligaments. As these enhancers were not solely accessible in the ligament cluster, enhancer specificity likely arises through a combination of chromatin accessibility and cell-type-restricted transcription factors. The identification of ligament-specific enhancers also highlights distinct gene regulatory programs for ligaments versus tendons, reflected by differential motif enrichment in enhancers. Some differential motifs include MEIS for ligaments and DLX for tendons, consistent with observed expression of *meis1b* in mesenchyme associated with the major jaw-stabilizing ligament mc-op. A recent study has also shown jaw cartilage defects and tendon disorganization in zebrafish *meis1b* mutants (Psutkova et al., 2025). Future experiments will be needed to determine how distinct types and combinations of motifs contribute to ligament- and tendon-specific enhancer activity. It will also be important to functionally validate enhancer activity on a larger scale by epigenetic profiling of histone marks, transgenic assays and mutagenesis in the endogenous context (Gasperini et al., 2020).

Previous work has mainly focused on the role of mechanical forces induced by muscle contraction in tendon remodeling (Subramanian et al., 2018), but how ligaments remodel as the skeleton changes (e.g. due to new bone addition) has remained less explored. By analyzing *scxa:mCherry* and osteoblast-associated transgenic lines, we identified a cell type intermediate between ligament and bone that may have a role in ligament remodeling. Initially, the major jaw-supporting ligament, mc-op, connects the jaw joint-associated bone with the op bone. However, another bone, the iop, later arises between these two bones. Remodeling of the mc-op ligament into separate ra-iop and iop-op ligaments coincides with ligament cells in the middle ossifying, thus splitting the initial ligament into two regions

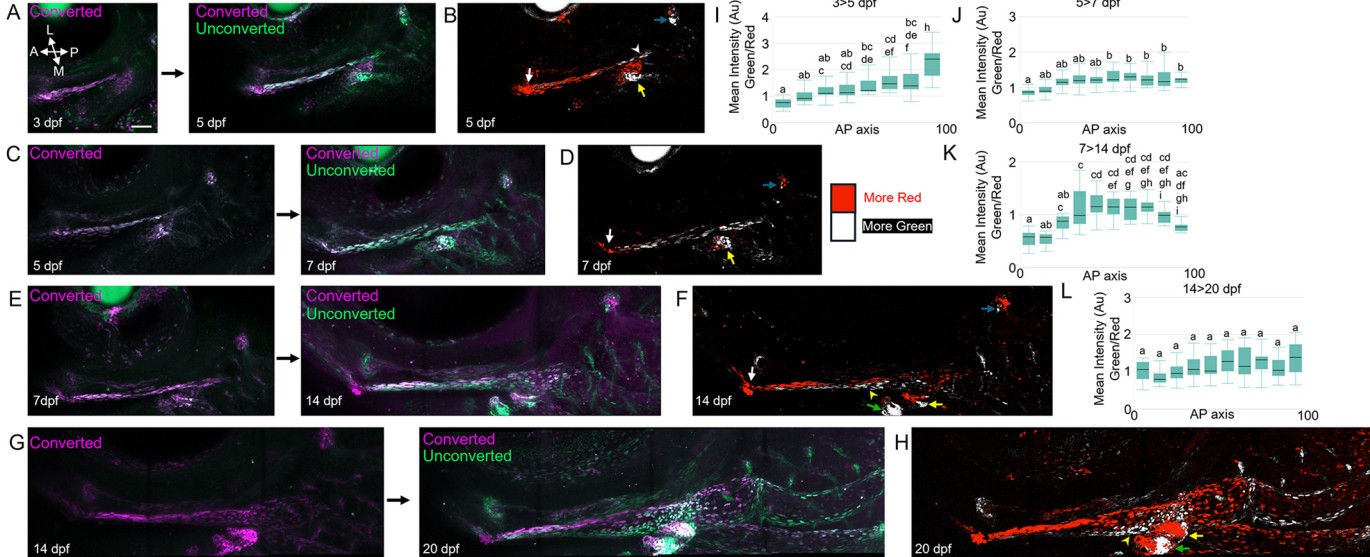

**Fig. 7. Lineage tracing of the major jaw joint-associated ligament.** (A,C,E,G) Confocal projections of the zebrafish face in lateral view show *thbs4a:nls:EOS* immediately after photoconversion (left) and then following the chase period (right). 3-5 dpf (A; *n*=13); 5-7 dpf (C; *n*=15); 7-14 dpf (E; *n*=8); 14-20 dpf (G; *n*=10). Scale bar: 50 µm. Anterior (A), posterior (P), lateral (L) and medial (M) axes are shown. (B,D,F,H) Cells with higher red/green ratio are pseudocolored red and those with higher green/red ratio are pseudocolored white. White arrows indicate the lack of new ligament addition at the jaw joint; white arrowhead in B indicates early posterior mc-op ligament addition; yellow arrowheads indicate medial growth of mc-iop ligament. Directional growth of the hs-op (blue arrows), ch-br1 (yellow arrows) and ch-br2 (green arrows) ligaments is also shown. (I-L) Box and whisker plots show differences in the green to red mean fluorescence intensity ratios (Au, arbitrary units) of segmented regions along the anterior-posterior lengths (AP, 0-100) of the mc-op/mc-iop ligament at different stages. Letters indicate statistical categories using one-way ANOVA and Tukey HSD test (segments not sharing letters are statistically different). Boxes represent the first and third quartiles, black lines the median, and the top and bottom bars the maximum and minimum values. See Fig. S6 for details of segmentation.

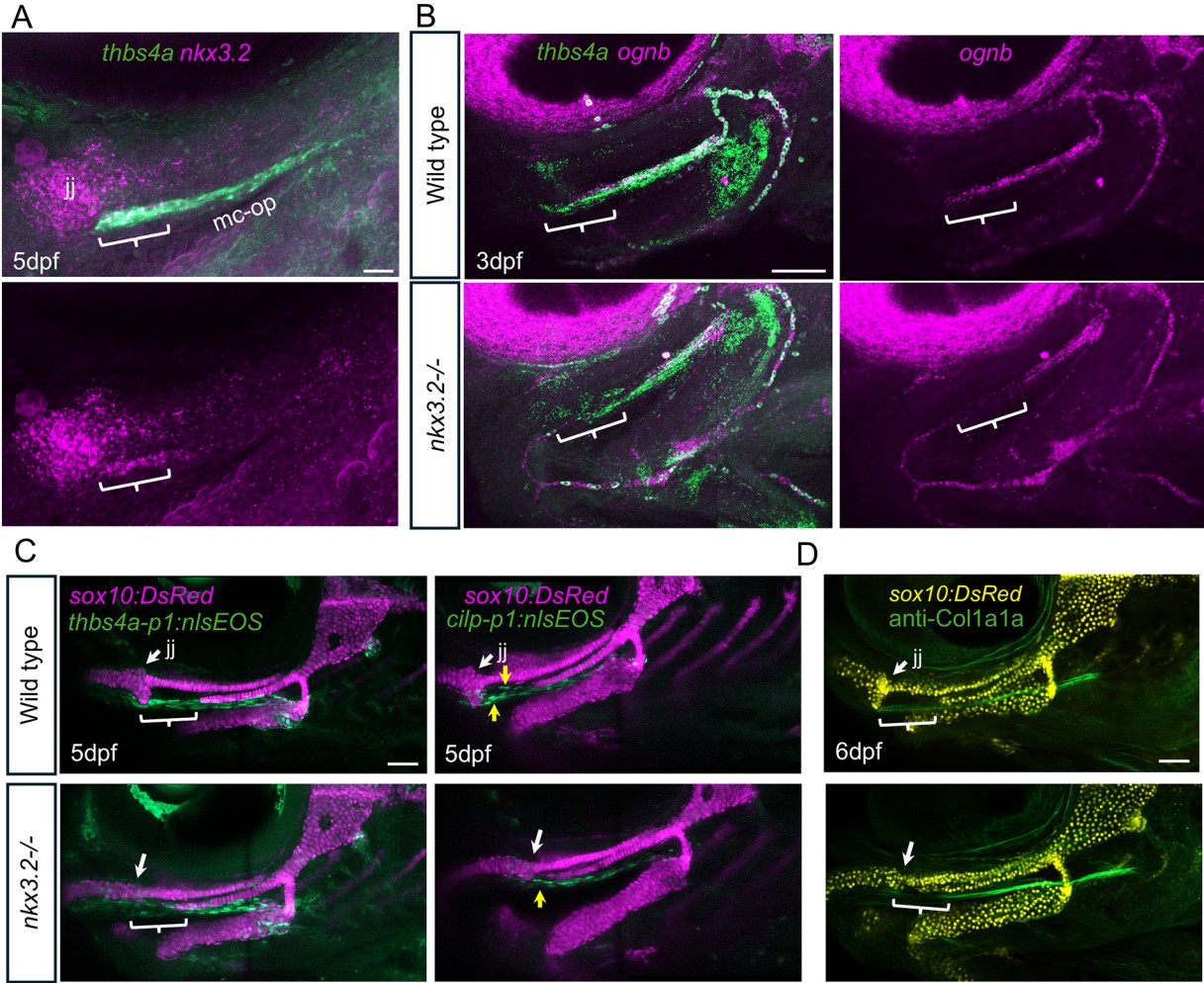

**Fig. 8. Requirement of *nkx3.2* for the anterior-most domain of the mc-op ligament.** (A) RNAscope *in situ* hybridization of a lateral view of the jaw joint (jj) shows co-expression of *nkx3.2* and *thbs4a* in the anterior region (brackets) of the mc-op ligament (*n*=3). (B) RNAscope *in situ* hybridization shows co-expression of *thbs4a* and *ognb* in the mc-op of controls and loss of anterior domain expression (brackets) in *nkx3.2*$^{-/-}$ mutants (wild type, *n*=5; *nkx3.2*$^{-/-}$, *n*=4). (C) Confocal projections show reduction of ligamentocyte *thbs4a-p1:nlsEOS* expression (brackets; wild type, *n*=3; *nkx3.2*$^{-/-}$, *n*=3) and dysmorphic periligament *cilp-p1:nlsEOS* expression (yellow arrows; wild type, *n*=3; *nkx3.2*$^{-/-}$, *n*=3) in the anterior domain of mc-op. *sox10:DsRed* labels cartilage for reference. White arrows indicate jaw joints in wild type and presumptive positions of missing jaw joint in mutants. (D) Col1a1a protein is reduced in the anterior domain (brackets) of mc-op of *nkx3.2*$^{-/-}$ mutants (wild type, *n*=4; *nkx3.2*$^{-/-}$, *n*=2). White arrows indicate jaw joints in wild type and presumptive positions of missing jaw joint in mutants. Scale bars: 20 µm in A; 50 µm in B-D.

that both connect to the iop through cells of mixed ligament and bone identity. These connections may be analogous to the fibrous enthesis, a transition zone between ligaments and bones (Wang et al., 2013). Similar enthesis-like cells were found to line the opercular series of intramembranous bones that support the gill cover. Last, we found that the midline ch-SH tendons form a sesamoid bone at their fusion point, reflected by initiation of osteoblast transgene activity in *scxa: mCherry*$^{+}$ cells and mineralization. This sesamoid bone may be similar to mammalian sesamoid bones that arise from bi-potent *Scx*$^{+}$/*Sox9*$^{+}$ cells in the limbs (Eyal et al., 2019). Thus, developmental ossification of tendons and ligaments is a conserved feature of vertebrates reflected by cells with mixed identities. This may help not only to connect ligaments to bones but also to facilitate normal remodeling of attachment sites, as well as underlying the propensity of tendons and ligaments to pathologically mineralize in congenital anomalies such as Eagle Syndrome (Camarda et al., 1989) and heterotopic ossifications that accompany injuries (Meyers et al., 2019).

Our lineage-tracing and mutant analysis also show distinct developmental mechanisms patterning different regions of an individual ligament. The mc-op ligament grows in the first 3 weeks by preferential addition of new ligament cells to posterior then medial regions, yet the anterior end connecting to the jaw joint is specified as early as 3 dpf. Formation of this anterior region selectively requires the function of Nkx3.2, which has more general roles in jaw joint specification (Miller et al., 2003; Smeeton et al., 2021). While expression of *nkx3.2* in anterior mc-op ligament cells is consistent with a cell-autonomous role in specifying this region of the ligament, we cannot rule out that loss of the anterior ligament domain could alternatively be due to absence of the jaw joint and ra bone. A number of previous studies in mice (reviewed by Bobzin et al., 2021) support that the bodies of limb tendons and ligaments may be specified by different mechanisms from their attachment sites. Our findings suggest that regional patterning genes, such as the jaw joint factor Nkx3.2, may serve to specify attachment sites in the face.

Our work also identified various subtypes of mesenchyme associated with tendons and ligaments that may function as progenitors and/or provide structural support. We identified two

distinct types of perichondrium, one of which was characterized by expression of *spon2b*, *scxa*, and *thbs4b*, and may represent a tendon and ligament progenitor. In support of this, a *six2a* enhancer drove perichondrium expression, and *Six2* has been shown to be an early gene in tendon differentiation in the distal limbs of mouse (Huang et al., 2015; Liu et al., 2015). We also identified a *cilp*[+] periligament cluster, although we were unable to establish through gene expression and transgenic testing whether this cluster also includes peritendon cells. As we did not observe contribution of *cilp_p1:nlsEOS*[+] periligament cells to *scxa:mCherry*[+] ligament cells based on transgene co-expression, periligament cells may function primarily as a sheath to support the ligament and not as the major progenitor source for ligament growth. We also uncovered a population of loosely distributed mesenchyme that may represent another potential source of progenitors for ligaments. These mesenchyme cells had preferential expression of *meis1b* and *alx4a*, with a previous study showing that *alx4a* marks a common progenitor for ligament and bone cells in zebrafish (Nichols et al., 2016). Future lineage tracing will be required to establish the extent to which these perichondral and mesenchymal subpopulations contribute to the formation and continued growth of tendons and ligaments during craniofacial development.

## MATERIALS AND METHODS
### Vertebrate animals
The University of Southern California Institutional Animal Care and Use Committee approved all zebrafish experiments (Protocol 20771). Zebrafish were maintained at 28.5°C under standard conditions. Published lines include *Tg(sox10:DsRedExpress)*[el10] (Das and Crump, 2012), *Tg(scxa:mCherry)*[fb301] (McGurk et al., 2017), *Tg(sp7:EGFP)*[b1212] (DeLaurier et al., 2010), *Tg(Hsa.RUNX2-Mmu.Fos:GFP)*[zf259] (Knopf et al., 2011), *nkx3.2*[el802] (Miyashita et al., 2020) and *TgBAC(col2a1a:EGFP-CAAX)*[el483] (Askary et al., 2015). The following transgenic lines were generated for this study: *Tg(cilp_p1-Mmu.E1b:NLS-EOS,cryaa:Cerulean)*[el1035], *Tg(thbs4a_p1-Mmu.E1b:NLS-EOS,cryaa:Cerulean)*[el1034], *Tg(scxa_p3-Mmu.E1b:GFP,cryaa:Cerulean)*[el1037], *Tg(fgfrl1a_p1-Mmu.E1b:GFP,cryaa:Cerulean)*[el1041], *Tg(meox1_p1-Mmu.E1b:GFP,cryaa:Cerulean)*[el1040], *Tg(six2a_p2-Mmu.E1b:GFP,cryaa:Cerulean)*[el1038], *Tg(sparc_p1-Mmu.E1b:NLS-EOS,cryaa:Cerulean)*[el1042] and *Tg(ucmaa_p2-Mmu.E1b:GFP,cryaa:Cerulean)*[el1043]. Addition founders included in Fig. S5 are *Tg(cilp_p1-Mmu.E1b:NLS-EOS,cryaa:Cerulean)*[el1044], *Tg(thbs4a_p1-Mmu.E1b:GFP,cryaa:Cerulean)*[el912], *Tg(scxa_p3-Mmu.E1b:GFP,cryaa:Cerulean)*[el1048], *Tg(fgfrl1a_p1-Mmu.E1b:GFP,cryaa:Cerulean)*[el1045], *Tg(meox1_p1-Mmu.E1b:GFP,cryaa:Cerulean)*[el1047], *Tg(six2a_p2-Mmu.E1b:GFP,cryaa:Cerulean)*[el1046] and *Tg(sparc_p1-Mmu.E1b:GFP,cryaa:Cerulean)*[el843].

### Generation of single-cell sequencing datasets
From *scxa:mCherry* zebrafish, we dissected and pooled 120 heads each for the 5 dpf scRNA-seq and multiome, 110 heads for the 5 dpf snATAC-seq and 80 heads for the 14 dpf scRNA-seq experiment. Heads were partitioned into Eppendorf tubes on ice (12 heads per tube for 5 dpf and 8 heads per tube for 14 dpf) and washed twice in Ringer's solution, followed by enzymatic dissociation in pre-warmed 20 mg/ml Collagenase D (from stock of 400 mg/ml Collagenase D in HBSS), and 0.25% trypsin and 1 mM EDTA (pH 8.0) in PBS at 28.5°C for 1 to 1.5 h with vigorous pipetting every 10 min. The reaction was stopped with 6 mM $CaCl_2$ and 30% fetal bovine serum (FBS) in PBS solution, and cells were collected by centrifugation at 400 *g* for 5 min at 4°C. Cells from each stage were pooled into one tube after being resuspended and rinsed twice with 1% FBS, 0.8 mM $CaCl_2$, 50 U/ml penicillin and 0.05 mg/ml streptomycin in Leibovitz Medium solution and passed through a cell strainer. Cells were incubated with DAPI and subjected to fluorescence-activated cell sorting (FACS). We collected ~100,000 *scxa:mCherry*[+] live cells (based on exclusion of DAPI) into a PBS solution with 0.04% BSA. Cells from each experiment were processed for nuclei isolation following 10X Genomics low cell count protocol

(protocol CG000169) for snATAC-seq and Multiome, or used directly for scRNA-seq. Nuclei were imaged on a confocal microscope for absence of blebbing or abnormal morphology prior to proceeding.

### Processing of single-cell sequencing datasets
Cell and nucleus partitioning, barcoding and library synthesis were performed following protocols of 10X Genomics for snMultiome (Chromium Next GEM Single Cell Multiome ATAC [+] Gene Expression, protocol CG000338), scRNA-seq (Single Cell 3' Reagent Kits v2, protocol CG00052) and snATAC-seq (Chromium Next GEM Single Cell ATAC Reagent, protocol CG000209). Libraries were sequenced on the Illumina NextSeq/HiSeq platform to reach a depth of 60,000 and 45,800 reads per cell for 5 and 14 dpf RNA libraries, and 10,600 reads per nuclei for the 5 dpf ATAC library. The 5 dpf multiome had a depth of 9000 reads per nuclei for the ATAC library and 736 genes per nuclei for the GEX library. Libraries were aligned against the reference genome built with the genomic sequence of GRCz11, the gene annotation of GRCz11.104, and the motif matrix of JASPAR2020; the mCherry sequence is built in as an extra chromosome. The accessible peaks of ATAC libraries were called and refined by Snaptools and SnapATAC as described by Fabian et al. (2022) to account for peaks contributed from cell clusters with smaller numbers. To calculate the cell-by-gene activity matrices of snATAC data, we used the gmat generated by SnapATAC as anchors to impute scRNA expression from corresponding libraries of the same developmental time point through anchoring functions in Seurat v5.1.0. All single-cell datasets were integrated based on the cell-by-gene matrices of scRNA and snMultiome datasets, as well as the cell-by-gene-activity matrices of snATAC datasets, through the integration functions in the Seurat package. The integrated datasets were used for further dimensional reduction by PCA and visualization by UMAP. For motif enrichment, CallPeaks from Signac package v1.14 was used to extract accessible peaks from tendon and ligament clusters. HOMER v4.11.1 was used for de novo motif discovery against random zebrafish genomic sequence. For the Motif Enrichment Assay using Signac package v1.14, top motifs for accessible peaks within clusters 2, 7 and 8 were identified relative to 40,000 randomly selected peaks from the tendon and ligament subset.

### RNA *in situ* hybridization
Fish embryos at each stage were euthanized with tricaine and fixed with 4% PFA overnight. Embryos were then dehydrated using 25%, 50%, 75% and 100% methanol in PBS with 0.1% Tween 20 (PBST20) for 10 min each and then stored at −20°C. Whole-mount in situ hybridizations were performed using the ACD Bio RNAscope or Molecular Instruments HCR technology. For RNAscope, the embryos were rehydrated using the reverse series of methanol PBST20 solutions, then washed in TBST20 and 1% BSA (TBST20-BSA) solution for 10 min. The samples were then transferred to a 95°C 1× target retrieval solution for 15 min. After incubation, embryos were immediately placed into TBST20-BSA for 1 min, 100% methanol for 1 min, and then back into TBST20-BSA. Protease plus solution was then added to the embryos, which were incubated at 40°C for 10 min (3 dpf stage), 15 min (5 dpf stage) or 30 min (14 dpf stage). Embryos were then rinsed with probe diluent and incubated with probes in channels 2, 3 or 4, diluted 1:50 in either probe 1 solution or probe diluent, at 40°C for 3 h. Following probe incubation, embryos were washed and placed in 5×SSC overnight. The next day, embryos were washed and placed in amplification solutions 1 through 3, with the first two for 30 min and the last for 15 min, with two intervening 5-min washes each. Following amplification, embryos were placed in HPR hybridization solutions matching the channel number for 15 min, washed and placed in TSA solution plus fluorophore for 30 min. Samples were then washed, followed by the HRP blocker solution for 15 min. The HRP hybridization step was then repeated for each probe used. Before imaging, samples were stored in 50% glycerol/PBS solution with DAPI. RNAscope probes include the following: Dr-spon2b-1175671-C3/4, Dr-thbs4a-812151-C1, Dr-cilp-1300411-C3, Dr-cilp2-1175681-C4, Dr-thbs4b-812161-C4, Dr-mkxa-1165631-C3, Dr-tnmd-564481-C1, Dr-angptl2b-1256461-C4, Dr-scxa-564451-C1/2, Dr-meis1b-01-1211871-C2, Dr-sox9a-543491-C1/3, Dr-hyal4-1000301-C4, Dr-nkx3.2-826281-C2, Dr-tnn-1175661-C2/C3 and Dr-ognb-1286381-C2. For HCR in situ hybridization, embryos were rehydrated in the same way as in the RNAscope protocol, then washed twice in PBST20 for

5 min each. Samples were placed in pre-chilled 100% Acetone for 12 min at −20°C, then washed three times for 5 min with PBST20. The samples were then incubated with 10 mg/ml of ProK diluted in PBST20 for 15, 30 or 45 min for 3, 5 and 14 dpf embryos, respectively. This was followed by three 5-min washes with PBST20, then fixation in 4% PFA for 20 min. The samples were washed five times with PBST20 for 5 min each, leading into the pre-hybridization step 2 of the Molecular Instruments whole-mount RNAFISH Zebrafish protocol (MI-Protocol-RNAFISH-Zebrafish-Rev9). HCR probes for *thbs4a*, *cilp*, *ognb*, *ecrg4a* and *ucmaa* were custom generated by the company.

### Whole-mount immunofluorescence and bone staining

Fish were fixed overnight at 4°C with 4% PFA and then washed with PBS for 5 min. Depending on age, samples were washed for 2-5 min with deionized $H_20$ and then placed at −20°C with 100% acetone for 7-15 min. Samples were rinsed for 1 min in deionized H20 and then rehydrated with PBS for 5 min. After rehydration, samples were incubated 15 min with PBDTx (5 ml 10×PBS, 0.5 g BSA, 500 µl DMSO, three drops of 1 M NaOH and 250 µl of 20% TritonX in a total volume of 50 ml), followed by a 3-h blocking step in PBDTx with 2% goat serum (PBDT-GS) at room temperature. The primary antibody was diluted in PBDT-GS, incubated overnight at 4°C, followed by three 20-min washes with PBDTx. The secondary antibodies and DAPI were diluted with PBDT-GS and incubated at room temperature for 5 h in the dark. Fish were then washed three times for 5 min each with PBS containing 1% Triton X and stored at 4°C in the dark in PBS until imaged. Antibodies include rabbit anti-mCherry (1:200, Novus Biologicals NBP2-25157), chicken anti-GFP (1:200, Abcam ab 13970), Alexa Fluor 488 goat anti-chicken (1:300, Thermo Fisher A-11039), Alexa Fluor 647 phalloidin (Thermo Fisher A22287), rabbit anti-Col1a1a (1:100, Genetex GTX133063) and goat anti-rabbit Alexa Fluor 568 (1:300, Thermo Fisher A11036). For live bone staining, fish were treated with Calcein Green at 1 mg/10 ml in embryo media (EM) in the dark for 30 min, followed by three washes for 5 min with EM.

### Generation of transgenic lines

Sequences corresponding to open chromatin regions were generated as gBlocks by Integrated DNA Technologies (see Table S2 for genomic coordinates and sequences) and cloned using the In-Fusion Snap Assembly kit (Takara Bio, 638945) into NLS-EOS or EGFP expression Tol2 vectors (Kawakami, 2007), with the E1b minimal promoter and cryaa:Cerulean as a selectable marker (blue lens). Embryos at the one-cell stage were injected with a mixture of 30 ng/µl of plasmid, 30 ng/µl Tol2 transposase RNA and Phenol Red. Injected embryos were screened between 3 and 5 dpf for mCerulean and either EGFP or NLS-EOS expression. Embryos with EGFP or NLS-EOS expression were raised to sexual maturity and crossed with the *scxa:mCherry* line to establish founders. For *thbs4a_p1* and *sparc_p1* enhancers, we identified independent nlsEOS and GFP lines with similar ligament expression. For *cilp-p1:nlsEOS*, two independent founders displayed periligament expression. For *six2a_p2:GFP*, 4/4 founders displayed perichondrium expression. For *meox1-p1:GFP*, 3/3 founders displayed perichondrium expression. For *scxa-p3:GFP*, one founder displayed periligament expression, with the other displaying stronger chondrocyte and weaker periligament expression. For *ucmaa-p1:GFP*, we identified only a single founder that showed highly specific mc-op ligament expression. For *fgfrl1a-p1:GFP*, one founder displayed periligament and hyoid joint perichondrium expression and a second founder displayed much weaker hyoid joint perichondrium expression.

### Imaging

Samples were mounted on coverslips in 0.2% agarose and imaged on a Zeiss LSM 800 confocal microscope using the ZEN Blue software with 10×, 20× or 40× objectives. Images were taken as three-dimensional stacks and shown as maximum intensity projections or individual slices as indicated. For larger areas, multiple tiles were stitched together using ZEN. For photoconversion of the *thbs4a_p1:nls:EOS* transgenic line at 3 dpf, we exposed one side of the face with a 405 nM laser for 3 min at 100% power using a 10× objective on a Zeiss LSM800 confocal microscope. For photoconversion of 5 dpf fish, we exposed larvae in clear plastic dishes with a UV flashlight for 1 h. For juvenile-stage fish, we used a UV nail curing device for 30 min for 7 dpf fish and 1 h for 14 dpf fish. We then used confocal imaging to confirm efficient green to red photoconversion. Images were adjusted for brightness in Adobe Photoshop or PowerPoint, with care taken to process images equally within experiments.

### Statistics

Except where indicated, $n \geq 3$ experiments were performed. For nls:EOS measurement, stitched CZI files from ZEN were converted to AMI files using the Amaris Converter. Within the Amaris imaging software, each image was cropped and segmented into 8 (3-5 dpf) or 10 (5-7, 7-14, 14-20 dpf) regions along the length of the mc-iop ligament. A surface mask was then generated in Amaris, and the mean fluorescence intensity for the green and red channels was calculated for each segment. The ratio of green to red mean fluorescence intensity was calculated and used to determine the minimum, median, maximum, and upper and lower quartiles for each segmented region, generating a box and whisker plot using Draxlr software. The Astata website was utilized to calculate a one-way ANOVA and Tukey HSD multivariate test to determine significant differences in the mean fluorescence intensity ratio between segments. To visualize cells with more red or more green in a binary fashion, ImageJ was used to first convert each channel to 32-bit and then invert the colors. The red channel was then divided by the green channel. A color look-up table was then modified to pseudocolor higher red cells red and higher green cells white. We also identified background pixels in the look-up table and set these to black, followed by adjustments for brightness and contrast.

### Acknowledgements

We thank Megan Matsutani and Maya Lujan for fish care, Seth Ruffins at the USC Optical Imaging Core for aid in image analysis, and David Ruble at the CHLA Sequencing Core for sequencing the datasets.

### Competing interests

The authors declare no competing or financial interests.

### Author contributions

Conceptualization: R.R.R., H.-J.C., J.G.C.; Data curation: R.R.R.; Formal analysis: R.R.R., A.B., K.-C.T., C.L.M., P.K.N., A.S., T.F.S.; Funding acquisition: T.F.S., A.E.M., J.G.C.; Investigation: R.R.R., A.B., H.-J.C.; Methodology: R.R.R., A.B., H.-J.C.; Resources: K.E., L.T., D.S., J.S.; Software: K.-C.T.; Supervision: T.F.S., A.E.M., J.G.C.; Writing – original draft: R.R.R., J.G.C.; Writing – review & editing: P.K.N., R.R.R., A.S., T.F.S., A.E.M., J.G.C.

### Funding

This work was supported by the National Institutes of Health (R35DE027550 to J.G.C.; R35DE034346 to A.E.M; R01DE033893 to J.G.C. and A.E.M.; DP2DE032725 to J.S.; and R01DE30565, R01AR67797 and R01DE13828 to T.F.S.). T.F.S. was also supported by the National Science Foundation (MCB2028424), N.P. by a National Science Foundation Simons Center for Multiscale Cell Fate fellowship, K.-C.T. by a California Institute for Regenerative Medicine Scholars EDUC4 training fellowship, and A.B. by Dean matching funds for a National Institute of Child Health and Human Development training program (T32 HD060549). Open Access funding provided by the University of Southern California. Deposited in PMC for immediate release.

### Data and resource availability

Single-cell datasets have been deposited in GEO under accession number GSE314322. All other relevant data and details of resources can be found within the article and its supplementary information.

### Peer review history

The peer review history is available online at https://journals.biologists.com/dev/lookup/doi/10.1242/dev.205045.reviewer-comments.pdf

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
