## [Peer Review File · Development (Cambridge, England)]

Gene regulatory programs underlying diversification of facial ligaments and tendons in zebrafish

Ryan R. Roberts, Arshia Bhojwani, Kuo-Chang Tseng, Kelsey Elliott, Hung-Jhen Chen, Lauren Teubner, Desmarie Sherwood, Joanna Smeeton, Cameron L. Miller, Pavan K. Nayak, Arul Subramanian, Thomas F. Schilling, Amy E. Merrill and J. Gage Crump
DOI: 10.1242/dev.205045

Editor: Steve Wilson

Review timeline

Original submission:	19 June 2025
Editorial decision:	11 August 2025
First revision received:	7 November 2025
Editorial decision:	15 December 2025
Second revision received:	19 December 2025
Accepted:	31 December 2025

Original submission

First decision letter

MS ID#: dev.205045

MS TITLE: Gene regulatory programs underlying diversification of facial ligaments and tendons in zebrafish

AUTHORS: Ryan Roberts; Arshia Bhojwani; Kuo-Chang Tseng; Kelsey Elliott; Hung-Jhen Chen; Lauren Teubner; Desmarie Sherwood; Joanna Smeeton; Cameron Miller; Pavan Nayak; Arul Subramanian; Thomas F Schilling; Amy E Merrill; J Gage Crump

Dear Gage,

I have now received all the referees' reports on the above manuscript, and have reached a decision. The referees' comments are appended below, or you can access them online: please go to: *****

As you will see, the referees express considerable interest in your work, but have some suggestions for improving the manuscript. If you are able to revise the manuscript along the lines suggested, I will be happy receive a revised version of the manuscript. Please also note that Development will normally permit only one round of major revision. If it would be helpful, you are welcome to contact us to discuss your revision in greater detail. Please send us a point-by-point response indicating your plans for addressing the referees' comments, and we will look over this and provide further guidance.

Please attend to all of the reviewers' comments and ensure that you clearly highlight all changes made in the revised manuscript. Please avoid using 'Tracked changes' in Word files as these are lost in PDF conversion. I should be grateful if you would also provide a point-by-point response detailing how you have dealt with the points raised by the reviewers in the 'Response to Reviewers' box. If you do not agree with any of their criticisms or suggestions please explain clearly why this is so.

Reviewer 1

Advance summary and potential significance to field

Roberts et al. describe the anatomy and corresponding molecular heterogeneity of facial ligaments and tendons in zebrafish. Using a combination of single cell genomics techniques, the authors present specific markers for distinct cellular subtypes, in particular identifying a subset of ligaments and tendons that appear to have osteogenic potential. They identify genomic regulatory elements that can drive gene expression that can distinguish tendons and ligaments, and through motif analysis suggest potential transcriptional regulators that may underlie cellular differences. Using photoconversion of the fluorescent protein Eos, they provide evidence for differential growth of the major jaw associated ligament. In addition they provide evidence of differential regulation of regions within the ligament, with deficits of ligament integration with the jaw joint in *nkx3.2* mutants. Overall the work is clearly presented, and will provide an important anatomical resource and molecular atlas for future studies of musculoskeletal development.

Comments for the author

I only have a few suggestions for authors to address:

Detailed methods for nlsEos photoconversion and imaging are lacking. What were imaging parameters? How were masks created? It would be good to represent data as red/green ratio as this could indicate old cells as well as new. It is somewhat disappointing that quantitative data are not provided to support the qualitative observations presented.

Figure 5. It is not clear how the green boxes in A correspond to tested enhancers.

Reviewer 2*Advance summary and potential significance to field*

In their manuscript "Gene regulatory programs underlying diversification of the facial ligaments and tendons in zebrafish" Roberts and colleagues use single cell multiomics to characterize the diversity of *scxa*-positive lineages in the zebrafish head. They first characterize the distribution of these lineages at 5 and 14 dpf and the time course of tendon and ligament development from 3 to 20 dpf. They perform snRNA/ATAC-seq, snATAC-seq and scRNA-seq at 5 dpf and scRNA-seq at 14 dpf. Following initial clustering of the 5 dpf data, they were reclustered using *col2a1a*, *sp7* and *thbs4b* to obtain a connective tissue subset. They obtained 16 clusters from this analysis and validated a set of these via fluorescent in situ hybridization. They then used the ATAC-seq data to identify enriched RNA motifs in the ligament, tendon and ossifying tendon/ligament clusters. They found that *Scx* was the most enriched motif in all 3 clusters and while there was overlap, each cluster had a distinct set of enriched motifs. They test 8 differentially accessible regions of DNA for transgenic activity and validate expression in appropriate tissues. They go on to characterize the development of one ligament, *Mc-OpL*, using one of these lines, *thbs4a_p1:nlsEOS*. They then examine the role of *nkx3.2* in specification of the anterior region of this ligament. The results with *nkx3.2* expression are interesting in that they demonstrate a molecular code in the ligament that matches that of the attachment site. The single cell sequencing data provides important and new knowledge regarding the development of the soft connective tissues in the head. The authors generate transgenic resources that are likely to be useful to the zebrafish community. The analyses in Figures 7 and 8, however, are difficult to interpret, seem preliminary in nature and do not integrate well into the manuscript.

Comments for the author

1) While the *scxa:mCherry* line is well characterized and appears to faithfully recapitulate endogenous express patterns, these analyses have not been performed at the single cell level. The authors should determine the percentage of cells in their full dataset and the connective tissue subset that are *scxa*-positive. The authors should also determine if any of their clusters are enriched in *scxa*-negative cells. From Figure 4B, this would appear to be the case. The concern being, are there cells in the dataset that were never truly *scxa*-positive? I realize that this may not

be an easy question to answer given the likely perdurance of mCherry protein relative to *scxa* mRNA. However, the extent to which this is possible should be characterized and discussed in the manuscript.

2) Figure 6 would benefit from a schematic to orient the reader, like shown in 4C.

3) The analyses in Figure 7 are not compelling. The authors state that the growth of the ligament is oriented in the posterior direction with no growth anteriorly and that the anterior region is specified earlier than the posterior. First, there is no clear definition of what is anterior versus posterior in these analyses. It appears that there are more green only cells in the posterior region of the ligament, however, there also appears to be more cells in general in the posterior region. There are green only cells in the anterior of the ligament. A better comparison would seem to be the proportion of green only cells in the anterior vs posterior. The authors' analyses do not provide compelling evidence that the anterior of the ligament is specified earlier than the posterior. Are there any ligament markers that are expressed anteriorly prior to being expressed posteriorly? Is there information in the scRNA-seq data that supports this claim?

4) The analyses with *nkx3.2* are difficult to interpret. As the authors note, the jaw joint is absent in *nkx3.2* mutants. Does the ligament form an attachment somewhere in the mutant? The loss of the jaw joint makes it difficult to disentangle a direct effect of *nkx3.2* versus an indirect effect of the loss of the integration site. The complete loss of *ognb* expression in the anterior ligament is the most compelling result as the alteration in shape of the other markers could readily be due to the loss of the retroarticular process. Here, again, can these analyses be tied into the single cell data?

5) It's unclear why the authors state that *thbs4b* is a marker of general mesenchyme (lines 156-157), when it's used as a tendon marker. Mesenchyme would be a broader cell type, which is unlikely to all be *thbs4b* positive, and would include bone and cartilage progenitors. Soft connective tissue might be a better term. Given that the sequencing is of *scxa*:mCherry-positive cells, defining any of the three markers (*col2a1a*, *sp7* or *thbs4b*) as a broad cell type is also a little confusing as the cells are actually double positive cells and would presumably be a small population of *col2a1a* and *sp7* expressing cells, in particular. It would clarify the issue if the words chondrocyte, osteoblast and mesenchyme were removed from the panel for Figure 3B. In the results section it could be made more clear that the authors are using this set of markers as a way to enrich for connective tissue.

Other minor concerns:

1) It's unclear why the authors used 3 different single cell analysis pipelines.

2) The title for Figure 8 should be changed. The jaw joint is not present in a *nkx3.2* mutant so the analyses are not testing the involvement of *nkx3.2* in the integration of the ligament with the jaw joint.

Reviewer 3

Advance summary and potential significance to field

In this study, the authors generated scRNA-seq and snATAC-seq data from *scxa*:mCherry+ cells to investigate the heterogeneity of facial tendons and ligaments. They identified clusters corresponding to tendon, ligament, periligament, perichondrium, and ossifying tendon/ligament. They also identified putative enhancers for each cluster and generated transgenic lines to test their activity during development. Overall, this is a highly interesting and resourceful study that provides a valuable database for researchers in the musculoskeletal field.

Comments for the author

Major points

Figure 1: It would be helpful to include the full names or codes for the tendon and ligament abbreviations in the figure legend.

In panel C, it is difficult to understand the morphology of several ventral tendons even with referencing the ventral cartoon in Fig S1C. It is difficult to make direct comparisons with illustrations in Subramanian et al. (co-submitted manuscript) in terms of this anatomy. For example, the way the ch-HH, ch-SH and HH-SH tendons are drawn in Fig 1C makes their morphology and connections to muscle/cartilage unclear. Is this depicting one continuous tendon made up of ch-HH, ch-SH and HH-SH domains (also unclear in Fig 4C)? There is no mention of HH-SH in Subramanian et al even though it is listed in Fig 2 as being present at 3 dpf. More importantly, this depiction is different from McGurk et al., 2017, (Fig 1 and Fig 6), which describes the ch-HH and ch-SH as having different relative dorsal-ventral locations to connect to their respective muscles. Subramanian et al also shows these tendons at different dorsal-ventral locations in their Fig 1B. The phalloidin stained HH and SH muscle in Fig 1A and Fig S1A of this manuscript appear to be at different dorsal-ventral locations. It would help to clarify this in the cartoons or provide 3D images or image slices to show relative positions of ch-SH, hh-SH, and HH-SH as the merged images are difficult to resolve depth and potential distinctions between these tendons. Also, in panel F, the label should be 'G' (not 'F').

Figure 8C: Given potential variability in transgene expression levels between fish, the reduction of *cilp_p1:nlsEOS* expression in the anterior domain of the mutant needs to be quantified.

Materials and Methods - Generation of Transgenic Lines: The specific expression of these enhancer elements is very exciting and could provide useful tools for future research. As such, only one founder for *ucmaa-p1:GFP*, *cilp-p1:nlsEOS*, and *fgfr1a-p1:GFP* is insufficient. Transgene expression can be affected by the integration site and copy number; therefore, multiple founders should be tested to confirm specificity of the enhancer element to the described regions.

Minor points

Figure 5H: Is the *RUNX2:GFP* transgene labeling cranial muscle? Please clarify the specificity of this transgenic line.

Figure 8C legend: "*sox:DsRed*" should be corrected to "*sox10:DsRed*".

Figure S1C: The morphology of HH-SH is not consistent with previous work as McGurk et al., 2017 (Fig 6) shows it as a rod-like structure. Please see above comments to clarify its morphology and dorsal-ventral location.

Figure S1E, F: These are described as lateral views, but they appear to be ventral. Please correct this in the figure legend.

Photoconversion Details: The manuscript is missing details about the photoconversion procedure—for example, the duration of UV exposure applied to the fish.

Text:

The authors describe cells with mixed tendon and osteoblast identity at tendon attachments to intramembranous bone in the context of fibrocartilaginous entheses. It seems relevant to also include a description of a possible relationship to fibrous entheses in mammals - in fibrous insertions, tendons attach directly to bone and migrate along the bone surface to accommodate longitudinal bone growth (PMID: 23109045).

There are different levels of functional definitions of enhancers throughout the literature (reviewed in Gasperini, 2020, PMID: 31988385). The most stringent definition requires demonstration that the region regulates the expression of a cis-located gene in its native context, in addition to reporter assays and marks of active enhancers. It is recommended that the authors include this in their discussion along with the limitations of their results in this context.

First revision

Author response to reviewers' comments

REVIEWER #1:

1.1 Detailed methods for nlsEos photoconversion and imaging are lacking. What were imaging parameters? How were masks created? It would be good to represent data as red/green ratio as this could indicate old cells as well as new. It is somewhat disappointing that quantitative data are not provided to support the qualitative observations presented.

Response: We thank the reviewer for this suggestion as it has allowed us to make more detailed insights into ligament cell addition. We added imaging parameters, including more detailed methods of photoconversion, in the *Imaging* sub-section of the Methods. We have also now increased our *n* numbers as well as performing additional photoconversion experiments at multiple stages: 3 to 5 dpf, 5 to 7 dpf, 7 to 14 dpf, and 14 to 20 dpf in revised Figure 7. As described in the revised Results (lines 273-298) and *Statistics* section of the Methods, we then divided the ligament along 8-10 segments along the anterior (jaw joint) to posterior (iop bone) axis and measured the red/green ratio in each segment with statistical significance between segments calculated using multivariate analysis (new Figure 7 and Figure S6). In new panels B,D,F,H of Figure 7 we also now pseudocolor cells with higher red/green ratio (red) versus higher green/red ratio (white) to better visualize ligament cell addition. This revealed statistically significant higher numbers of new cells (high green-red ratio, pseudocolored white) toward more posterior segments at all stages except 14-20 dpf, consistent with addition of new ligament cells during remodeling of the posterior attachment to the late-forming iop bone. From 14-20 dpf, new ligament cell addition then shifts toward more medial regions closer to the midline.

1.2 Figure 5. It is not clear how the green boxes in A correspond to tested enhancers.

Response: In legend to Figure 6A, we now clarify that “green boxes correspond to cell types displaying transgenic activity of tested enhancers linked to the listed genes”.

REVIEWER #2:

2.1 While the *scxa*:mCherry line is well characterized and appears to faithfully recapitulate endogenous express patterns, these analyses have not been performed at the single cell level. The authors should determine the percentage of cells in their full dataset and the connective tissue subset that are *scxa*-positive. The authors should also determine if any of their clusters are enriched in *scxa*-negative cells. From Figure 4B, this would appear to be the case. The concern being, are there cells in the dataset that were never truly *scxa*-positive? I realize that this may not be an easy question to answer given the likely perdurance of mCherry protein relative to *scxa* mRNA. However, the extent to which this is possible should be characterized and discussed in the manuscript.

Response: In new Figure S2E, we now show the percentage of cells with *scxa* expression per cluster for the different subsets. We discuss this in the Results section.

Lines 169-176: “When analyzed for *scxa* expression, all clusters in the connective tissue subset and tendon and ligament subset had variable numbers of *scxa*⁺ cells, while some epithelia and red blood cell clusters in the original combined datasets had no *scxa*⁺ cells suggesting recovery from FACS due to autofluorescence (Fig. S2E). In addition, similar proportion of *scxa*⁺ cells in some osteoblast and chondrocyte clusters compared to tendon and ligament clusters is consistent with these representing enthesis-like clusters intermediate between skeletal and soft connective tissue types.”

2.2 Figure 6 would benefit from a schematic to orient the reader, like shown in 4C.

Response: A schematic has been added as Figure 6B.

2.3 The analyses in Figure 7 are not compelling. The authors state that the growth of the ligament is oriented in the posterior direction with no growth anteriorly and that the anterior region is specified earlier than the posterior. First, there is no clear definition of what is anterior versus posterior in these analyses. It appears that there are more green only cells in the posterior region

of the ligament, however, there also appears to be more cells in general in the posterior region. There are green only cells in the anterior of the ligament. A better comparison would seem to be the proportion of green only cells in the anterior vs posterior. The authors' analyses do not provide compelling evidence that the anterior of the ligament is specified earlier than the posterior. Are there any ligament markers that are expressed anteriorly prior to being expressed posteriorly? Is there information in the scRNA-seq data that supports this claim?

Response: Reviewer 1 also had concerns with Figure 7 (see comment 1.1) and thus we have substantially redone the analysis with more stages, quantification of the red/green ratio, definition of anterior and posterior regions, and more nuanced discussion of the growth of the ligament. In revised Figure 7, we present additional photoconversion experiments from 3 to 5 dpf, 5 to 7 dpf, 7 to 14 dpf, and 14 to 20 dpf. As described in the revised Results (lines 273-298) and *Statistics* section of the Methods, we then divided the ligament along 8-10 segments along the anterior (jaw joint) to posterior (iop bone) axis and measured the red/green ratio in each segment with statistical significance between segments calculated using multivariate analysis (new Figure 7 and Figure S6). Note that we were not able to confidently segment all nuclei and hence we used overall red/green ratio per segment as a proxy for the number of red and green cells per segment. This analysis revealed higher numbers of new cells (high green-red ratio) toward more posterior segments at all stages except 14-20 dpf, consistent with greater addition of new ligament cells during remodeling of the posterior attachment to the late-forming iop bone. From 14-20 dpf, new ligament cell addition shifted to regions closer to the midline. We also better denote a small population of cells at the attachment to the jaw joint that remains red with little to no new green nlEOS production for each of the experimental time-courses, indicating that these cells form at or prior to the first photoconversion (3 dpf). In our scRNAseq data, we did not uncover clusters indicative of ligament cells at different stages of specification and hence do not have clear temporal markers to independently validate the nlEOS photoconversion results.

2.4 The analyses with *nkx3.2* are difficult to interpret. As the authors note, the jaw joint is absent in *nkx3.2* mutants. Does the ligament form an attachment somewhere in the mutant? The loss of the jaw joint makes it difficult to disentangle a direct effect of *nkx3.2* versus an indirect effect of the loss of the integration site. The complete loss of *ognb* expression in the anterior ligament is the most compelling result as the alteration in shape of the other markers could readily be due to the loss of the retroarticular process. Here, again, can these analyses be tied into the single cell data?

Response: We agree with the reviewer that we cannot rule out cell-autonomous versus non-cell-autonomous roles of *Nkx3.2* in specifying the anterior-most portion of the ligament that attaches to the jaw joint. Due to the loss of ligament gene expression in the anterior-most domain near where the jaw joint should have formed, we were unable to determine if the ligament forms an attachment somewhere in the mutant. Analysis of single-cell data was also uninformative as *nkx3.2* is expressed in only a small region of a single ligament, reflected by its expression in only a few cells in the ligament cluster. The in situ data in Figure 8A showing expression of *nkx3.2* in *thbs4a+* ligament cells near the jaw joint is therefore more compelling data that *Nkx3.2* could have a direct role in anterior ligament specification. We have now revised the Discussion to make clear that our data do not distinguish between direct and indirect effects on the anterior region of the mc-op ligament.

Lines 405-408: “While expression of *nkx3.2* in anterior ligament cells is consistent with a cell-autonomous role in specifying this portion of the ligament, we cannot rule out that loss of the anterior ligament domain could alternatively be due to indirect effects of loss of the jaw joint and ra bone.

2.5 It's unclear why the authors state that *thbs4b* is a marker of general mesenchyme (lines 156-157), when it's used as a tendon marker. Mesenchyme would be a broader cell type, which is unlikely to all be *thbs4b* positive, and would include bone and cartilage progenitors. Soft connective tissue might be a better term. Given that the sequencing is of *scxa:mCherry*-positive cells, defining any of the three markers (*col2a1a*, *sp7* or *thbs4b*) as a broad cell type is also a little confusing as the cells are actually double positive cells and would presumably be a small population of *col2a1a* and *sp7* expressing cells, in particular. It would clarify the issue if the words chondrocyte, osteoblast and mesenchyme were removed from the panel for Figure 3B. In the

results section it could be made more clear that the authors are using this set of markers as a way to enrich for connective tissue.

Response: We agree with the reviewer that “soft connective tissue” is a better term than “mesenchyme” to refer to *thbs4b*⁺ cells. We revised Figure 3B to refer to *thbs4b*⁺ “soft connective tissue”. However, we kept chondrocyte for *col2a1a* and osteoblast for *sp7* as we feel these are well established markers of these cell types and the cell type terms help the reader understand what types of cells are being marked. That said, we fully agree that the *scxa:mCherry* sort likely captured only a small proportion of chondrocytes and osteoblasts that co-express *scxa*. In the Results, we now better describe using these markers “to enrich for connective tissue type” and also clarify that the recovered osteoblasts and chondrocytes likely represent “enthesis-like clusters intermediate between skeletal and soft connective tissue types”.

2.6 It's unclear why the authors used 3 different single cell analysis pipelines.

Response: This reflects the fact that the project was performed over a period of 5 years. When we began, commercial single-cell multiome kits were not available and hence we performed RNA-seq and ATAC-seq experiments separately. When multiome kits became available, we then performed additional linked RNA-seq and ATAC-seq to better couple transcriptome and chromatin accessibility data in single cells.

2.7 The title for Figure 8 should be changed. The jaw joint is not present in a *nkx3.2* mutant so the analyses are not testing the involvement of *nkx3.2* in the integration of the ligament with the jaw joint.

Response: We fully agree and have changed the Figure 8 title to “Nkx3.2 is required for development of the anterior region of the mc-op ligament”.

REVIEWER #3:

3.1 Figure 1: It would be helpful to include the full names or codes for the tendon and ligament abbreviations in the figure legend. In panel C, it is difficult to understand the morphology of several ventral tendons even with referencing the ventral cartoon in Fig S1C. It is difficult to make direct comparisons with illustrations in Subramanian et al. (co-submitted manuscript) in terms of this anatomy. For example, the way the ch-HH, ch-SH and HH-SH tendons are drawn in Fig 1C makes their morphology and connections to muscle/cartilage unclear. Is this depicting one continuous tendon made up of ch-HH, ch-SH and HH-SH domains (also unclear in Fig 4C)? There is no mention of HH-SH in Subramanian et al even though it is listed in Fig 2 as being present at 3 dpf. More importantly, this depiction is different from McGurk et al., 2017, (Fig 1 and Fig 6), which describes the ch-HH and ch-SH as having different relative dorsal-ventral locations to connect to their respective muscles. Subramanian et al also shows these tendons at different dorsal-ventral locations in their Fig 1B. The phalloidin stained HH and SH muscle in Fig 1A and Fig S1A of this manuscript appear to be at different dorsal-ventral locations. It would help to clarify this in the cartoons or provide 3D images or image slices to show relative positions of ch-SH, hh-SH, and HH-SH as the merged images are difficult to resolve depth and potential distinctions between these tendons. Also, in panel F, the label should be 'G' (not 'F').

Response: We agree that the original confocal projections were not clear as regards to the distinct ventral tendons. After reevaluation of the data, in particular individual confocal slices, both we and the Subramanian paper have decided to rename the HH-SH tendon to HH-HA. We have also added a midline HH-HH tendon that was missing from our paper. In new panels Figure S1I-J, we provide individual slices at different medial and lateral depths (note that the zebrafish field uses “dorsal” and “ventral” instead to denote “proximal” and “distal”). These slices now clearly show that the bilateral ch-HH tendons, short HH-HH tendon, and rod-like midline HH-HA tendon are more medial, with the bilateral ch-SH tendons located more laterally and coming into close contact at their posterior attachments to the SH muscles. We also corrected Figure 1F to 1G.

3.2 Figure 8C: Given potential variability in transgene expression levels between fish, the reduction of *cilp_p1:nlsEOS* expression in the anterior domain of the mutant needs to be quantified.

Response: We attempted to quantify total expression levels but results were inconclusive. We have therefore modified the language to denote the dysmorphic nature of the domain in mutants rather than a reduction in expression levels. This qualitative change was consistent between 3 wild-type sibling controls (left) and 3 mutants (right). We have attached all examples here.

Lines 310-313: “We also observed reduced levels of the ligament transgene *thbs4a_p1:nlsEOS* and a dysmorphic domain (single layer instead of two) of the periligament transgene *cilp_p1:nlsEOS* in the anterior mc-op region at 5 dpf”

3.3 Materials and Methods - Generation of Transgenic Lines: The specific expression of these enhancer elements is very exciting and could provide useful tools for future research. As such, only one founder for *ucmaa-p1:GFP*, *cilp-p1:nlsEOS*, and *fgfr1a-p1:GFP* is insufficient. Transgene expression can be affected by the integration site and copy number; therefore, multiple founders should be tested to confirm specificity of the enhancer element to the described regions.

Response: We identified an additional founder for both the *cilp-p1:nlsEOS* and *fgfr1a-p1:GFP* lines that displayed similar expression patterns. We were unsuccessful in identifying a second founder line for *ucmaa:GFP*. We now clearly state this caveat in the Results (Lines 255-257): “We confirmed expression in multiple founder lines for each enhancer, with the exception of *ucmaa_p1:GFP* for which only a single founder was identified.”. We also now show representative images of independent transgenic founders in new Figure S5.

3.4 Figure 5H: Is the *RUNX2:GFP* transgene labeling cranial muscle? Please clarify the specificity of this transgenic line.

Response: The *RUNX2:GFP* line has been used extensively to label osteoblasts in zebrafish. We increased the gain to better visualize weaker expression in the ch-SH sesamoid bone, which led to off-target *RUNX2:GFP* muscle activity appearing in the image. Reducing the gain (image below), shows the normal documented expression of *RUNX2:GFP*. We now clearly denote this muscle expression with asterisks in the figure. We also note that *RUNX2:GFP* results are independently validated with *sp7:GFP* and Calcein Green staining of bone in Figure 5G-I. The off-target activity of *RUNX2:GFP* at higher gain settings in cranial muscle therefore does not affect our conclusion that a sesamoid bone forms within the HH-HA tendon.

3.5 Figure 8C legend: "sox:DsRed" should be corrected to "sox10:DsRed".

Response: Corrected.

3.6 Figure S1C: The morphology of HH-SH is not consistent with previous work as McGurk et al., 2017 (Fig 6) shows it as a rod-like structure. Please see above comments to clarify its morphology and dorsal-ventral location.

Response: We agree and now present individual confocal slices in new Figure S1I-J that clearly show HH-HA (revised name for HH-SH) as a midline rod-like structure.

3.7 Figure S1E, F: These are described as lateral views, but they appear to be ventral. Please correct this in the figure legend.

Response: Corrected.

3.8 Photoconversion Details: The manuscript is missing details about the photoconversion procedure—for example, the duration of UV exposure applied to the fish.

Response: We added imaging parameters, including more detailed methods of photoconversion, in the *Imaging* sub-section of the *Methods*. See also response to comment 1.1.

3.9 The authors describe cells with mixed tendon and osteoblast identity at tendon attachments to intramembranous bone in the context of fibrocartilaginous entheses. It seems relevant to also include a description of a possible relationship to fibrous entheses in mammals - in fibrous insertions, tendons attach directly to bone and migrate along the bone surface to accommodate longitudinal bone growth (PMID: 23109045).

Response: We thank the reviewer for altering us to this citation and the general concept of a fibrous entheses. We have now updated the Discussion accordingly: "These connections may be analogous to the fibrous entheses, a transition zone between ligaments and bones (Wang et al., 2013)."

3.10 There are different levels of functional definitions of enhancers throughout the literature (reviewed in Gasperini, 2020, PMID: 31988385). The most stringent definition requires demonstration that the region regulates the expression of a cis-located gene in its native context, in addition to reporter assays and marks of active enhancers. It is recommended that the authors include this in their discussion along with the limitations of their results in this context.

Response: We agree with the reviewer that the most stringent test of enhancers is regulation of cis-located genes in their native context, though testing enhancer requirements is often challenging due to enhancer redundancy and the presence of shadow enhancers. We have added this limitation to the Discussion and added the Gasperini, 2020 reference.

Lines 370-372: "It will also be important to functionally validate enhancer activity at larger scale by epigenetic profiling of histone marks, transgenic assays, and mutagenesis in the endogenous context (Gasperini et al., 2020)."

Second decision letter

MS ID#: dev.205045R1

MS TITLE: Gene regulatory programs underlying diversification of facial ligaments and tendons in zebrafish

AUTHORS: Ryan Roberts; Arshia Bhojwani; Kuo-Chang Tseng; Kelsey Elliott; Hung-Jhen Chen; Lauren Teubner; Desmarie Sherwood; Joanna Smeeton; Cameron Miller; Pavan Nayak; Arul Subramanian; Thomas F Schilling; Amy E Merrill; J Gage Crump

Dear Gage,

The referees are happy with your revisions and there are just a couple of very minor points to address before publication. Please attend to all of the reviewers' comments in your revised manuscript and detail them in your point-by-point response.

Reviewer 1*Advance summary and potential significance to field*

The authors have addressed my previous concerns and improved the manuscript.

Reviewer 2*Comments for the author*

The authors have addressed my previous concerns.

Reviewer 3*Advance summary and potential significance to field*

The authors have done a sufficient job addressing all my previous comments. Their manuscript will provide a rich resource for the developmental biology community and presents novel findings about diversification of connective tissues in the zebrafish.

There are a few very minor comments:

Figure 1G: The label still reads "F" in the figure image.

Figure 3: In the related text, it describes using thbs4b as a marker of "soft connective tissue" and further describes expression in the tendon and ligament clusters (scxa-high, thbs4b high) and mesenchyme (scxa-high, thbs4b low), but there is no marker(s) listed for the 5 connective tissue clusters. From Fig S3, it appears that most of the CT clusters express thbs4b. Could thbs4b be listed as a marker for the CT clusters in this statement (line 153)? It helps to clarify which cells were found within the "thbs4b+ soft connective tissue" group.

Also, related to the terminology change, Fig 3C has "thbsb+ mesenchyme clusters" in the figure legend and "mesenchyme" in the image, which should be changed to keep the terminology consistent.

Second revisionAuthor response to reviewers' comments

Reviewer 3: There are a few very minor comments:

Figure 1G: The label still reads "F" in the figure image.

- corrected

Figure 3: In the related text, it describes using *thbs4b* as a marker of "soft connective tissue" and further describes expression in the tendon and ligament clusters (*scxa*-high, *thbs4b* high) and mesenchyme (*scxa*-high, *thbs4b* low), but there is no marker(s) listed for the 5 connective tissue clusters. From Fig S3, it appears that most of the CT clusters express *thbs4b*. Could *thbs4b* be listed as a marker for the CT clusters in this statement (line 153)? It helps to clarify which cells were found within the "*thbs4b*+ soft connective tissue" group.

- We now clarify on line 153 that the 5 connective tissue clusters are *thbs4b*+

Also, related to the terminology change, Fig 3C has "*thbsb*+ mesenchyme clusters" in the figure legend and "mesenchyme" in the image, which should be changed to keep the terminology consistent.

- To keep the terminology consistent, we have changed "mesenchyme" to "*thbs4b*+ soft connective tissue clusters" in both Fig. 3C image and the accompanying legend

Third decision letter

MS ID#: dev.205045R2

MS TITLE: Gene regulatory programs underlying diversification of facial ligaments and tendons in zebrafish

AUTHORS: Ryan Roberts; Arshia Bhojwani; Kuo-Chang Tseng; Kelsey Elliott; Hung-Jhen Chen; Lauren Teubner; Desmarie Sherwood; Joanna Smeeton; Cameron Miller; Pavan Nayak; Arul Subramanian; Thomas F Schilling; Amy E Merrill; J Gage Crump

Dear Gage,

I am happy to tell you that your manuscript has been accepted for publication in Development, pending our standard publication integrity checks.